# Fetal biometry and amniotic fluid volume assessment end-to-end automation using Deep Learning

Saad Slimani [1,2] ✉, Salaheddine Hounka[3], Abdelhak Mahmoudi [1,4], Taha Rehah[1], Dalal Laoudiyi[2], Hanane Saadi[5], Amal Bouziyane[6], Amine Lamrissi[2], Mohamed Jalal[2], Said Bouhya[2], Mustapha Akiki[7], Youssef Bouyakhf[1], Bouabid Badaoui[8,9], Amina Radgui[3], Musa Mhlanga [10] & El Houssine Bouyakhf[1]

Fetal biometry and amniotic fluid volume assessments are two essential yet repetitive tasks in fetal ultrasound screening scans, aiding in the detection of potentially life-threatening conditions. However, these assessment methods can occasionally yield unreliable results. Advances in deep learning have opened up new avenues for automated measurements in fetal ultrasound, demonstrating human-level performance in various fetal ultrasound tasks. Nevertheless, the majority of these studies are retrospective in silico studies, with a limited number including African patients in their datasets. In this study we developed and prospectively assessed the performance of deep learning models for end-to-end automation of fetal biometry and amniotic fluid volume measurements. These models were trained using a newly constructed data-base of 172,293 de-identified Moroccan fetal ultrasound images, supplemented with publicly available datasets. the models were then tested on prospectively acquired video clips from 172 pregnant people forming a consecutive series gathered at four healthcare centers in Morocco. Our results demonstrate that the 95% limits of agreement between the models and practitioners for the studied measurements were narrower than the reported intra- and inter-observer variability among expert human sonographers for all the parameters under study. This means that these models could be deployed in clinical conditions, to alleviate time-consuming, repetitive tasks, and make fetal ultrasound more accessible in limited-resource environments.

Ultrasound (US) is a low-cost, non-invasive imaging modality that has been shown to independently reduce fetal mortality by up to 20%[1]. Yet, 99% of preventable fetal and maternal deaths occur in developing countries where access to fetal ultrasound is scarce and more than a third of operators have no training at all[2,3]. The WHO recommends at least one US examination for each pregnancy[4]; however, there is a shortage of physicians and sonographers able to perform this examination primarily in countries of the Global South[5]. These countries are not the only ones suffering from excessive and increasing fetal and maternal mortality. The USA ranks last amongst industrialized countries in terms of maternal mortality with notable ethnic differences: African-American women are three times more likely to die during pregnancy compared to non-Hispanic White women[6]. Thus, democratizing access to healthcare resources dedicated to fetal and maternal health, regardless of ethnicity, socioeconomic status, or geographic location, is a global healthcare priority.

Two vital and systematic assessments of all routine screening scans are fetal biometry (FB) and amniotic fluid volume (AFV). FB and AFV help detect and manage potential life-threatening conditions. FB is used to determine gestational age (GA), which is essential to guide therapeutic interventions in the case of pre-term labor or pre-eclampsia and detect pregnancy-related complications, such as fetal growth restriction (FGR). FGR, sometimes defined as the "failure of the fetus to meet its growth potential due to a pathological factor"[7], is responsible for 30% of all stillbirths and poor neonatal outcomes. Its diagnosis can rely solely on US FB assessment when abdominal circumference (AC) or estimated fetal weight (EFW) falls below the 3rd percentile[8,9]. AFV abnormalities are strongly associated with increased mortality in the case of low AFV (oligohydramnios)[10]. The single deepest pocket (SDP) method has been proven to be as reliable as the amniotic fluid index method (AFI) for AFV assessment but to cause fewer false positive diagnoses for oligohydramnios and, therefore, fewer unnecessary labor inductions.

FB coupled with AFV assessments are time-consuming, repetitive, and error-prone tasks, and several studies have stressed the need for quality audits to ensure measurement reproducibility and lower inter and intra-observer variability[11–13].

Advances in deep learning (DL) applied to medical imaging have sparked interest in its application to measurement automation in fetal ultrasound, with studies showcasing human-level performances of DL models in standard plane classification and segmentation[14–17]. Most of them are retrospective "in silico" studies conducted on Caucasian populations on fixed images, except for a few exceptions[18]. To the best of our knowledge, no African team has ever led a study on African patients to automate fetal ultrasound tasks using DL. We believe it to be of the utmost importance that the Global South should not simply import Artificial Intelligence (AI) breakthroughs in medicine from the Global North, but rather developed mindfully, and responsibly by researchers aware of the local constraints and characteristics[19].

Furthermore, no previous work has tried to develop an understandable approach to the automation of both tasks respecting the quality guidelines set forth by the International Society of Ultrasound in Obstetrics and Gynecology (ISUOG) for the FB workflow or allowing an end-to-end automation of the AFV assessment workflow, (see Related Methods below for extensive comparison with existing methods). Recent approaches using "blind" cine-loops represent a paradigm shift as they do seem to allow for the democratization of gestational age estimation and their use by minimally trained sonographers but fail to promote both the autonomy and education of the operators[18,20].

An end-to-end, understandable, FB and AFV assessment workflow automation could not only potentially alleviate practitioners' burden, increase ultrasound sensitivity and specificity, and even enable minimally trained healthcare workers to perform these measurements in resource-stranded environments but also empower them to learn how to perform these tasks in the absence of such a tool.

Here we train and test DL models, in clinical conditions, to effectively automate these two tasks from standardized free-hand videos.

## Results

### Data

To train and test DL models meant to fully automate FB and AFV assessment, the models were trained on a newly built database of 172,293 de-identified fetal ultrasound images from 12,356 US exams done in six health centers in two different cities in Morocco between 2015 and 2021. In addition, publicly available datasets were used with the following ultrasound machines: General Electric's Voluson E6, E8, E10, S8, and S10, and Aloka[17].

Within the collected data, 30,249 2D standard biometry planes of the abdomen, brain, and femur were preprocessed to extract pixelated annotations. The preprocessing allows it to recognize the region of interest, then to detect the pixelated colored calipers and circumferences to generate the ground truth masks for the abdomen and brain. The femur was annotated directly by the annotators. Optical character recognition (OCR) techniques were used to extract the information and acronyms about the standard plan and the biometric measurements as measured by the doctors during screening. In the end, the annotators validated the extracted masks, standard plan, and measurements (see supplementary information file for details).

In total, fifteen human annotators (ranging from medical students to Radiology and Obstetrics professors) participated in the annotation process using our bespoke annotation platform based on the open-source tool Label Studio version 1.3.0[21] that we adapted to our needs. Each annotation indicated the type of standard plane (abdomen, brain, femur), a polygonal segmentation in the case of the femur, and some of the quality criteria associated with it as described by the ISUOG guidelines[22] (Table 1). The annotators made a further distinction between transthalamic, transcerebellar, and transventricular planes. Quality criteria such as the zoom (head, abdomen, femur occupying more than half of the image – caliper placement – the angle of the femur to the horizontal <45°) were omitted in the annotation process. Instead, their detection was automated through fetal structure segmentation: calculating the surface ratio of the structure to the whole image or the angle of the femur to the horizontal to determine conformity to the criteria described by Salomon et al.[23] (Table 1). That step was designed to ensure that the models select the best suitable plane on a given video loop, detecting the presence or absence of the quality criteria, and displaying them with the measurement, allowing an insight into the model's choice as well as a correction if necessary.

Images were also annotated according to the presence or absence of an AF pocket, defined as an in-utero fluid pocket free of fetal parts or the umbilical cord. In the case of the presence of the AF pocket, annotators were asked to segment it manually.

Figure 1 shows a summary of the amount of annotated data for the segmentation of the three biometric structures and their classification based on their own quality criteria, along with the number of individual measurements in the annotated data for the classification and segmentation of AF pockets.

### Segmentation and classification models

We assessed the performances of the segmentation and classification models on the retrospective test sets, comparing them to the experts'

**Table 1 | Criteria for score-based biometry plane assessment developed by Salomon et al.[23]**

| Cephalic | Abdominal | Femur |
|---|---|---|
| Symmetrical plane | Symmetrical plane | Both ends of bone clearly visible |
| Plane showing thalami | Plane showing stomach bubble | <45° to horizontal |
| Plane showing cavum septum pellucidi | Plane showing portal sinus | Femur occupying more than half of total image |
| Cerebellum not visible | Kidneys not visible | Calipers placed correctly |
| Head occupying more than half of total image | Abdomen occupying more than half of total image | |
| Calipers and dotted ellipse placed correctly | Caliper and dotted ellipse placed correctly | |

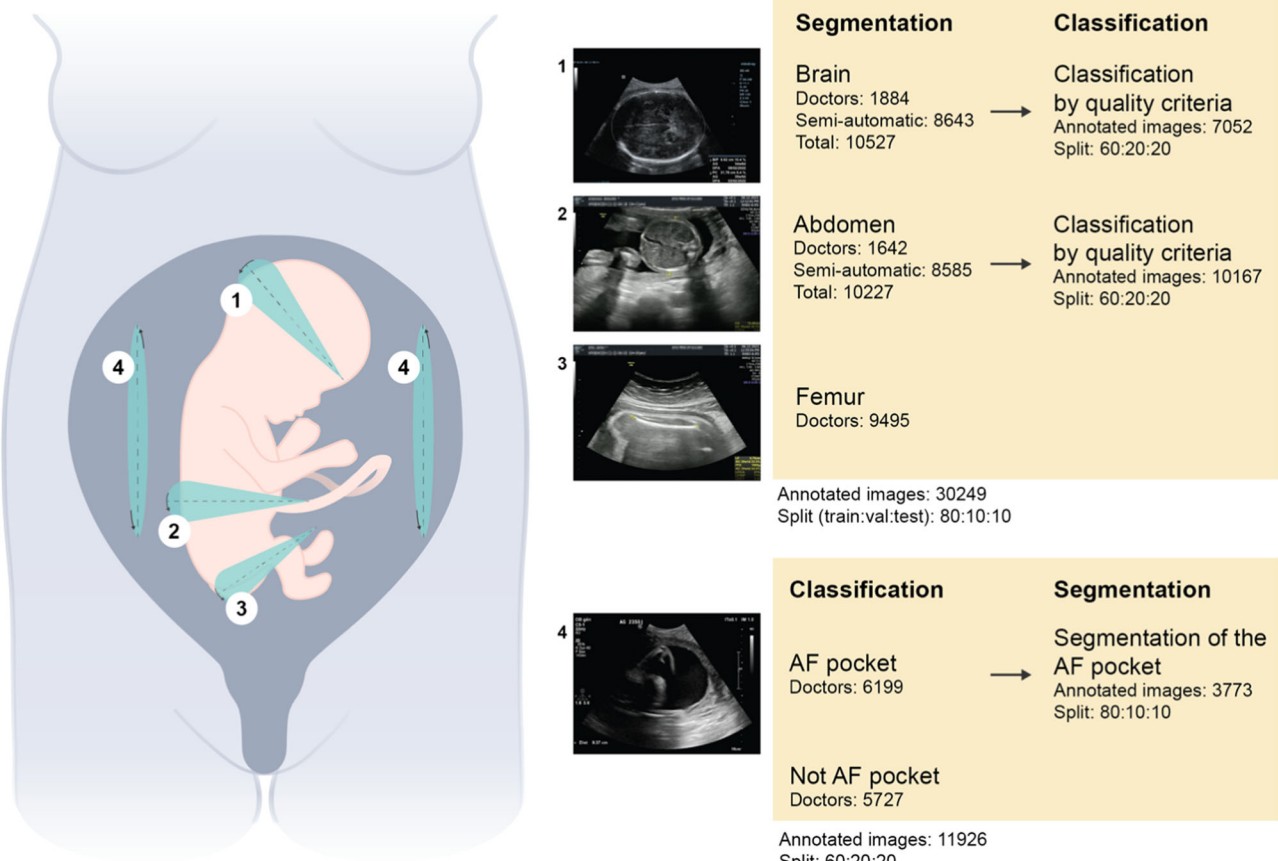

**Fig. 1 | Summary of the retrospective data used during the segmentation and classification tasks along with the volume of data used for training, validation, and testing.** 'Doctors' refer to physicians who prospectively and manually annotated standard planes. 'Semi-automatic' refers to the process of the standard plane and biometric measurement recognition using Optical Character Recognition, validated by a trained research technician. Helena Pinheiro https://www.hpinheiro.com/ created this illustration.

annotations for standard plane detection, quality criteria detection, brain, abdomen, and femur structure segmentation, and AF pocket detection and segmentation.

We used a set of Mask-R-CNN architectures for the standard plane detection and anatomical regions (brain, abdomen, and femur) segmentation[24]. Mask-R-CNN models are widely used in the state-of-the-art literature. However, the datasets used for training and testing the models were not sufficient to efficiently evaluate their performance and they have been used only for one single task of fetal biometry. For example, Al-Bander et al. and Moccia et al. adapted the Mask-R-CNN architecture to assess HC measurement using the HC18 dataset[25,26]. Mask-R-CNN models are generally used, for instance, segmentation. The idea is to train models that can detect two or more instances of an object in the images, for example, two femurs, two heads in the case of twins, or separable regions of the amniotic fluid region. Mask-R-CNN has a backbone architecture with a certain number of network depth features that are taken from a certain number of final convolutional layers on a certain training schedule. The notation used to nominate the architectures used in the experiment is the same used by detectron2[27]. For example, R_50_C5_3x is a Mask-R-CNN architecture with ResNet as a backbone. It has 50 depth features taken from the convolutional layer of the fifth stage and a training schedule of 3x, which means 1 iteration every 3 × 12 (36) epochs.

In our study, four Mask-R-CNN models were finetuned (R_101_C4_3x, R_101_DC5_3x, R_50_C4_3x, R_50_DC5_3x) for the segmentation of the brain, abdomen, and femur using 30249 annotated images (10527 brains, 10227 abdomens, and 9495 femurs). The R_50_DC5_3x model achieves the best performance with an average DICE score of 0.89 and an Intersection over Union (IoU) score of 0.82 versus 0.96 and 0.90 respectively, reported with the FUVAI model[14] (Fig. 2). The Segmentation of the brain region achieved the best performance with a DICE score of 0.95 and an IoU of 0.91.

For each biometry plane, classification models for quality criteria detection were assessed on the test set of the retrospective data (Fig. 3). Assessment of the quality of the standard biometry plane allows for better reproducibility of the AC measurement, we assessed 4 quality criteria (kidneys not visible (A_KN), plane showing portal sinus (A_PS), plane showing stomach bubble (A_SB), symmetrical plane (A_SYM)) leaving out the image zoom quality criteria that are the only one that is not qualitative and can be inferred directly from the abdomen segmentation. Based on three finetuned models (INCEPTIONV3, RESNET50V2, and VGG16), we obtained F1 scores of 0.81, 0.80, and 0.80, respectively, INCEPTIONV3 shows the best results for all the criteria with an average area under the curve (AUC) of 0.86, and an F1 score of 0.81. The results also show that the A_SB criterion is detected better compared to other criteria with an AUC of INCEPTIONV3 of 0.93.

For the classification of the brain plane, five quality criteria were assessed: cerebellum not visible (B_CB), plane showing cavum septum pellucidity (B_CS), plane showing posterior horn of lateral ventricles (B_PVV), symmetrical plane (B_SYM) and plane showing thalami (B_TH). Similarly, the 3 classification models were finetuned for this task. They show very similar results with an average AUC of 0.83. We obtained F1 scores of 0.66, 0.62, and 0.62 for INCEPTIONV3,

## Segmentation scores for each biometry plane

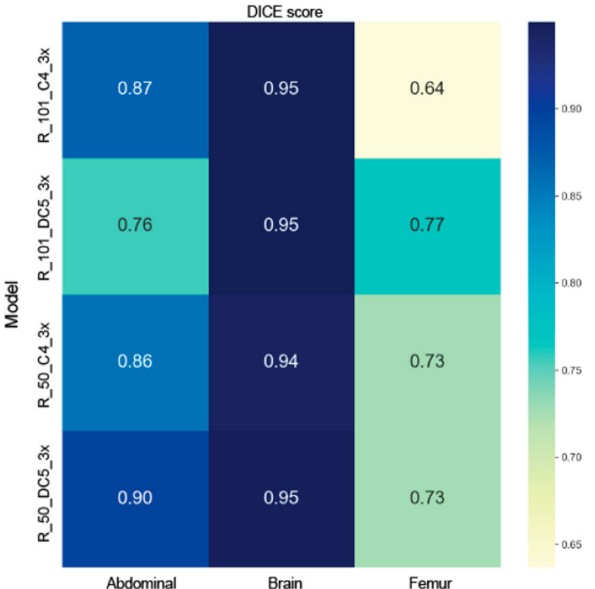
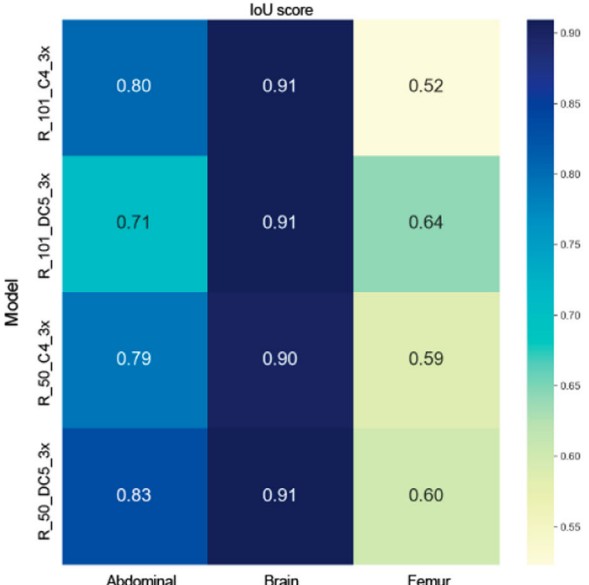

**Fig. 2 | Overall DICE and IoU scores of four versions of four finetuned Mask-R-CNN models for the segmentation of the biometric planes.** These models ('R_101_C4_3x', 'R_101_DC5_3x', 'R_50_C4_3x', 'R_50_DC5_3x') were trained for the segmentation of the abdominal, femoral, and brain planes on the retrospective test set. The bar plot shows the segmentation performances on the three biometric structures, and the heatmap plots show the DICE (left) and IoU (right) scores per structure. The R_50_DC5_3x model achieves the best performance with a DICE score of 0.89 and an IoU score of 0.82. The segmentation of the brain achieved the best performance with a DICE score of 0.95 and an IoU score of 0.91.

RESNET50V2, and VGG16, respectively. The results also show that the B_CB criterion is well detected compared to other criteria with an AUC of INCEPTIONV3 of 0.95 (Fig. 3).

For the femoral plane, the performances of the model designed to detect if both ends of the femur are clearly visible were poor as inter-observer variability was high in the training set; thus, it was not used for image quality scoring. For the femoral plane on the prospective part of the study, the size, subsequent femur to image sizes ratio, and angle of the femur were directly obtained from the femur segmentation stage and kept as the only quality criteria.

For the AF pocket classification, we compared the finetuned models (INCEPTIONV3, RESNET50V2, and VGG16) on the retrospective test set (Fig. 4). The results show almost equivalent AUC scores of 0.89, and F1 scores of 0.81, 0.78, and 0.80, respectively. Similarly, we compared seven finetuned Mask-R-CNN models ('R_101_C4_3x', 'R_101_DC5_3x', 'R_101_FPN_3x', 'R_50_C4_3x', 'R_50_DC5_3x', 'R_50_FPN_3x', 'X_101_32x8d_FPN_3x') for the segmentation of the AF pocket region (Fig. 4). These models were trained on 3773 images manually annotated with polygons by the experts out of 6199 ones that were annotated as containing AF pocket from the total number of 11926 images.

The results show that 'X_101_32x8d_FPN_3x' achieved the best performance with a DICE score of 0.78 and an IoU of 0.71 versus a DICE of 0.877 for the state-of-the-art model by Cho et al.[28] who tested the model on only 125 images.

In this retrospective study, we adopted the finetuned R_50_DC5_3x model for the segmentation of the brain, abdomen, and femur, the finetuned INCEPTIONV3 models for the quality criteria and the AF pocket detection, and the finetuned X_101_32x8d_FPN_3x model for the AF pocket segmentation. These models will then be evaluated based on the prospectively acquired data.

### Models performance on the prospective evaluation

From October 2021 to April 2022, 172 patients with singleton pregnancies were included in our prospective study, the average age of the participants was 30.38 years (minimum: 18, maximum: 44, standard deviation: 6.05), most of the included patients did not have any comorbidity (87%), ten patients lived with diabetes mellitus (6%) and three lived with a hypertensive disorder (2%). 34 patients (20%) were nulliparous, 47 (27%) were primiparous, and 91 (53%) were multiparous. Multiple pregnancies were not an exclusion criterion, and patients were included even in the case of partially complete examinations. However, duplicates and patients without an image or cine-loop available or no corresponding ground truth measurement obtained were excluded (Fig. 5). In total, the study gathered: 142 different cine-loops containing a femoral plane; 144 containing an abdominal plane; 123 containing a cephalic plane; and 90 containing AF pockets. GA estimation from first-trimester ultrasound from crown-rump length measurement (CRL) was unavailable in almost all cases (see supplementary data 1).

Overall, the mean GA estimated by the operators was of 30 weeks and 3.13 days ± 6 weeks and 3.1 days (range: 15 weeks and 2 days – 41 weeks and 2 days), the mean measured HC, BPD, AC, FL, EFW, and SDP were respectively of 26.37 ± 5.88 cm (range: 11.29–34.71 cm), 7.41 ± 1.72 cm (range: 3.09–10.07 cm), 23.98 ± 6.58 cm (range: 8.95–38.18 cm), 5.28 ± 1.44 cm (range: 1.52–7.86 cm), 1606.78 ± 957.56 g (range: 108.81–3783.86 g) and 5.25 ± 2.22 cm (range: 2.15–17.37 cm).

The US machines and healthcare centers from which the prospective data differed from those of the retrospective data were retained. Three of the four centers where the prospective part of the study was conducted did not participate in the retrospective data collection. Several US machines used in the prospective testing were not present in the retrospective data as well: Mindray DC 40 and Resona 6, Philips Medical Systems Affinity 50 W, 70 G, and GE Voluson P8.

When possible, EFW and GA were computed from all measurements using the recommended Hadlock and Intergrowth formulae[29,30] and all necessary measurements performed by the doctors with the corresponding available cine-loops.

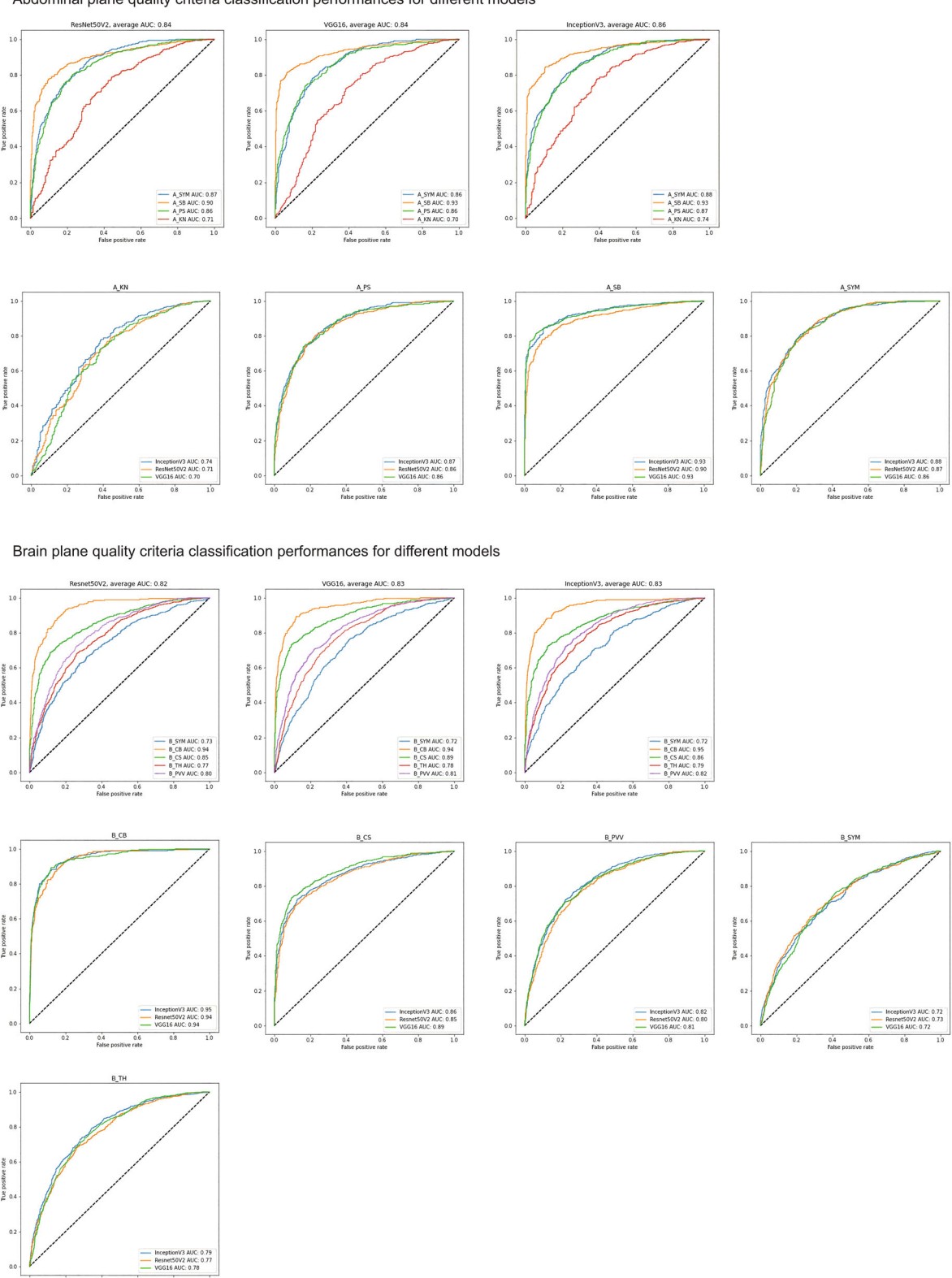

**Fig. 3 | Comparison of the receiver operating characteristics (ROC) curves of three finetuned models (INCEPTIONV3, RESNET50V2, and VGG16) for the brain and abdominal planes classification using their respective quality criteria.** Overall, the three models show similar results for the cephalic plane quality criteria, and INCEPTIONV3 shows the best results for the abdominal criteria with an average AUC of 0.86. Legends: (kidneys not visible (A_KN), portal sinus visible (A_PS), stomach bubble visible (A_SB), abdominal plane symmetry (A_SYM), brain plane symmetry (B_SYM), cerebellum not visible (B_CB), cavum septum visible (B_CS), posterior horn of lateral ventricle visible (B_PVV) and thalami visible (B_TH)) on the retrospective test set. The top row shows the classification per model, and the bottom row shows the results per quality criteria.

AF pocket detection

a)

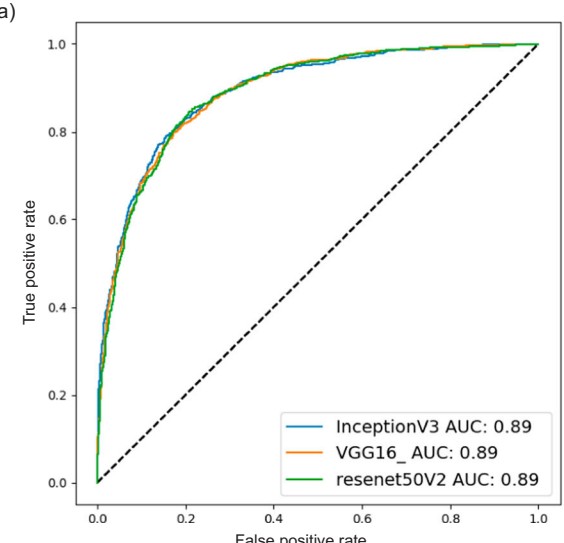

AF pocket segmentation

b)

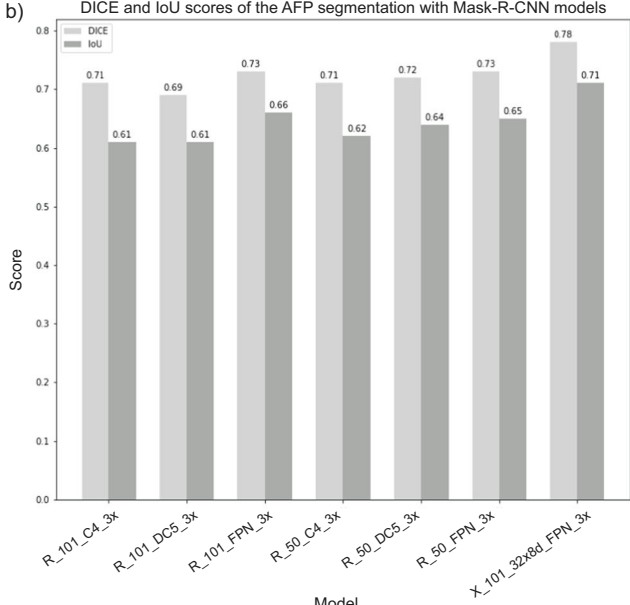

**Fig. 4 | ROC curves and bar plot of the Amniotic Fluid Pocket (AFP) classification and segmentation performances on the retrospective test set. a** AUC of three finetuned models for the AFP classification. The results show equivalent AUC scores of 0.89. **b** DICE and IoU scores of seven finetuned Mask-R-CNN models for the AFP segmentation. The results show that 'X_101_32x8d_FPN_3x' achieved the best performance with a DICE score of 0.78 and an IoU of 0.71.

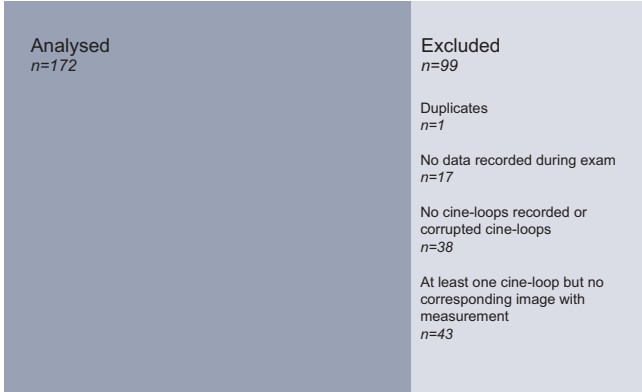

**Fig. 5 | Study flow chart.** 172 patients were analyzed and 99 were excluded mainly because the measurements corresponding to cine-loops were not saved by the operators ($n = 43$), followed by the absence of valid cine-loops recorded ($n = 38$), no data recorded during the study ($n = 17$) and duplicates ($n = 1$).

Hadlock formula for EFW estimation[29]:

$$\log_{10} EFW = 1.335 - 0.0034 \times AC \times FL + 0.0316 \times BPD + 0.0457 \times AC + 0.1623 \times FL$$

Intergrowth recommended formula for GA estimation >14 weeks[30]:

$$\log_e(GA) = 0.03243 \times (\log_e HC)^2 + 0.001644 \times FL \times \log_e HC + 3.1813$$

The models segmented each relevant anatomical region and then extracted the planes with the highest composite score, including the quality score according to the ISUOG subjective quality criteria, the zooming of the image inferred from the anatomical segmentation to total image ratio, and the confidence of the model's prediction (Fig. 6).

The models were able to extract measurements from all the videos containing standard biometry planes. The 95% limits of agreement expressed in percentage using the Bland-Altman method were ±0.54% for HC, ±3.74% for BPD, ±0.14 % for AC, ±3.11% for FL, ±1.45% for GA, ±2.42% for EFW, and ±16.96 % for SDP. All percentages found are narrower than reported inter and intra-observer limits of agreements among sonographers (HC: ±3.0%, AC: ±5.3%, FL: ±6.6% for intra-observer difference, and HC: ±4.9%, AC: ±8.8%, FL: ±11.1 for inter-observer difference)[31] (Fig. 7). Visual assessment of the Bland-Altman plots shows random error for every parameter, the variability increasing with the size of the parameter. However, our results also show constant bias for SDP and FL, the predicted measurements for both parameters being consistently greater than those of the physicians.

This over-expectation of the femur segmentation by the model can be mitigated by reviewing the images manually. By selecting images with abnormal results, we found that the model often selected planes showcasing strictly horizontal femurs and that the predicted calipers were placed to avoid the grand trochanter in accordance with measurement guidelines. Participating physicians did not always follow these guidelines (Fig. 8)[30].

As for the SDP discrepancy, it appears as though the model actually detected deeper pockets that were not selected or measured by the clinician. However, the model's failure can also be explained by a slight angulation of the probe from 90° results in a larger anteroposterior pocket diameter at the time of examination which will be construed as the SDP by our approach (Fig. 8).

The ICC for each measurement was high (>0.9 for all parameters apart from SDP), showing excellent reliability of the measurements: AC = 0.982, HC = 0.987, BPD = 0.975, FL = 0.945, GA = 0.978, EFW = 0.9713, SDP = 0.692.

The MAE for each biometric parameter was 0.67 ± 0.69 cm for HC, 0.33 ± 0.22 cm for BPD, 0.27 ± 0.40 cm for FL, 0.91 ± 0.81 cm for AC, 9.85 ± 14.36 days for GA, 147.18 ± 177.97 g for the EFW and 1.46 ± 1.10 cm for SDP (Table 2).

The FUVAI model is the closest one to our approach for end-to-end automated biometric assessment from cine-loops and showed similar performances to those of trained sonographers[14] (see related methods for more comparisons with existing methods).

We computed the MAE of each parameter using the open-source FUVAI model developed by Płotka et al.[14] and compared them with our approach (Table 2).

It showed inferior MAE compared with our approach for every biometric parameter except for BPD, we hypothesize that this is due to the fact that HC and BPD are measured from the same mask with our approach. However, it was often the case that operators took two distinct images to measure each, hence the difference in performance. We also note that our approach was able to correctly detect the entirety of the corresponding biometry plane while FUVAI failed to do so.

The MAE between the predicted SDP and the measured SDP was also lower than the one reported by Cho et al.[28] with their state-of-the-art model for AF pocket segmentation: AF-net (1.46 cm with our approach vs. 2.666 cm for Cho et al.[28] on a retrospectively annotated dataset).

There were no cases of oligohydramnios in the prospective set and 7 cases (7.07%) of polyhydramnios. The sensitivity and specificity of the models at detecting polyhydramnios were 86.6%, and 85.7%, respectively, when comparing them to the experts' estimation.

The models' estimated biometric parameters were computed during the prospective phase of the study at the earliest time after each examination was complete. No adverse effect was reported during the entirety of this study. Participants were not compensated for their participation in the study.

## Discussion

In this study, we were able to create a successful end-to-end method to automatically estimate FB and AFV from ultrasound cine-loops using the ISUOG quality criteria for standard biometry planes, with results that were similar to those of expert operators. These two tasks are part of the six fundamental items listed by the ISUOG in the recently updated practice guideline for the routine mid-trimester scan[31]. They

allow early detection of life-threatening conditions such as FGR, oligohydramnios, and polyhydramnios that are associated with increases in the risk of fetal mortality by respectively 19, 5, and 3 fold[32–34].

At present, sonographers must navigate through a series of steps to capture the correct anatomical plane. This involves adjusting the probe angle and position to meet the appropriate quality standards, freezing the image, and accurately positioning the calipers. This entire procedure, excluding the assessment of amniotic fluid volume, typically involves an average of 12 steps. The assessment of amniotic fluid volume adds additional steps, as it requires identifying the largest AF pocket, freezing the screen, and then positioning the calipers[31]. The method we have developed simplifies this process significantly, enabling reliable biometric tasks to be performed by merely recording the relevant structures. This reduces the process to just three steps: initiating the recording, sweeping through the desired anatomical structure, and concluding the recording. If incorporated into the clinical workflow, this approach has the potential to decrease scanning time, alleviate the workload of sonographers, and ensure the optimal reliability of the measurements taken.

In addition to streamlining the fetal ultrasound, our approach has the potential to serve as an aid to the sonographer in training in the recognition of the quality criteria of each standard biometry plane and gaining independence in their practice.

If deployed in conditions where healthcare workers receive minimal ultrasound training, as is often the case in countries of the Global South, this approach has the potential to help patients receive accurate fetal biometry assessment, gestational age estimation, fetal weight estimation, as well as amniotic fluid volume assessment which in turn are key to diagnosing fetal growth and amniotic fluid volume abnormalities.

Indeed, the 95% limits of agreement expressed in percentage between the models' measurements and the doctors' measurements for AC, HC, FL, and SDP were narrower than both reported intra- and inter-observer variability for human expert sonographers[13,35]. The differences between the US machines, the operators, and the healthcare facilities in the retrospective and the prospective data indicate that the developed models are generalizable. Furthermore, our deterministic

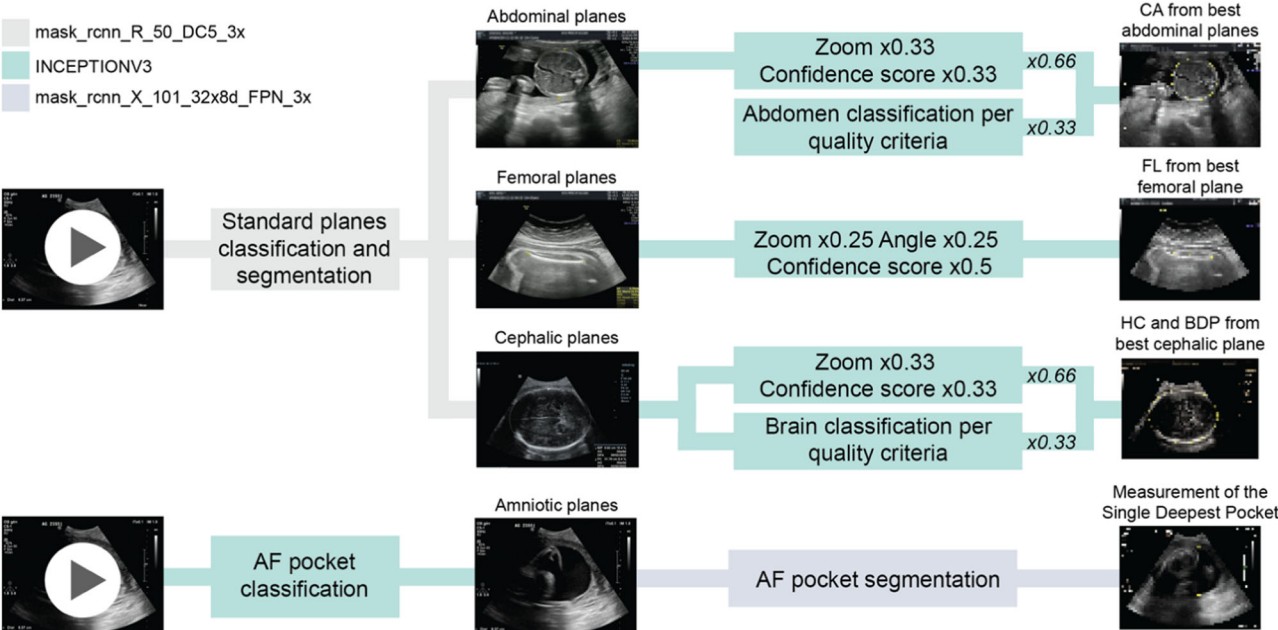

**Fig. 6 | Flow chart of the end-to-end automated extraction of biometric parameters from ultrasound cine-loops.** In every cine-loop, all standard biometry planes are detected, the relevant anatomical structures are segmented, then the quality criteria of each plane are assessed and the highest-scoring plane is selected. There is no quality assessment in the case of the AF volume assessment, the AF pocket with the larger depth is selected from the cine-loop.

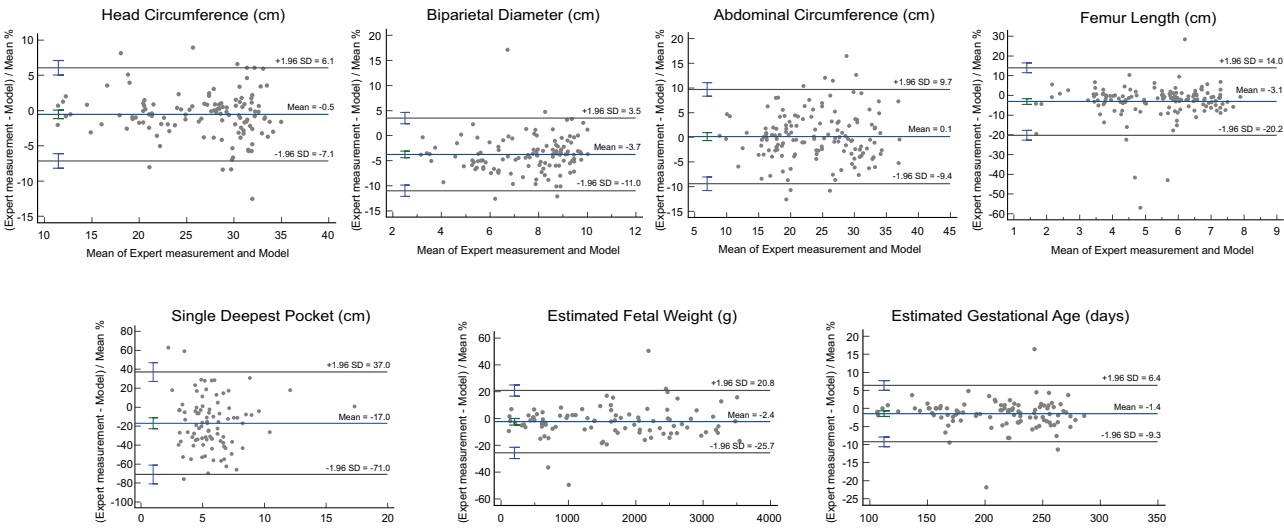

**Fig. 7 | Bland-Altman plots showing the variability between the models and the doctors HC, FL, EGA, AC, AF (single deepest pocket), BPD, and EFW.** The plots are derived from n = 172 biologically independent samples. The three horizontal lines in each plot represent the mean difference (middle line) and the limits of agreement (upper and lower lines), which are defined as the mean difference plus and minus 1.96 times the standard deviation of the differences. The error bars represent standard deviation (SD). Source data are provided as a Source Data file.

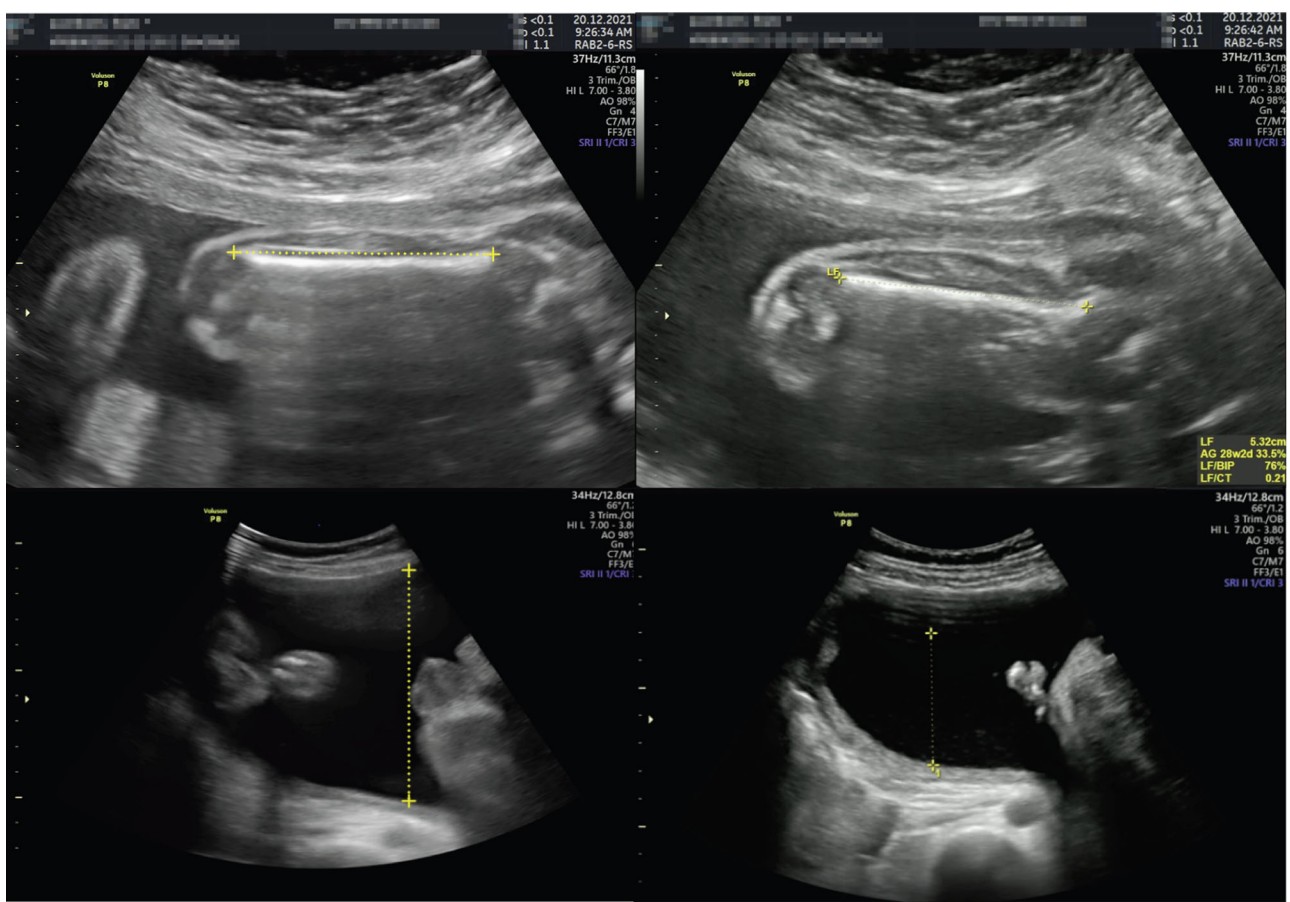

**Fig. 8 | Examples of larger predicted (left) than measured (right) femur lengths (FL) and single deepest pockets (SDP) in the same study participants.** The femur selected by the model (left) was strictly horizontal, abiding by the quality criteria, as opposed to the femur the physician selected (right). The AF pocket, correctly measured by the model (left), is larger than the one measured by the physician due to an error in caliper placement by the human operator.

**Table 2 | Table comparing measurements performed by clinicians, our approach, and the FUVAI model**

| | HC | BPD | FL | CA | EFW | GA | SDP |
|---|---|---|---|---|---|---|---|
| MAE ± standard dev with our approach | 0.67 ± 0.69 cm | 0.33 ± 0.22 cm | 0.27 ± 0.40 cm | 0.91 ± 0.81 cm | 147.18 ± 177.97 g | 9.85 ± 14.36 days | 1.46 ± 1.10 cm |
| Correctly detected planes from cine-loops with our approach | 123 (100%) | 142 (100%) | 144 (100%) | NA | NA | NA | 90 (100%) |
| MAE ± standard dev FUVAI | 0.70 ± 0.67 cm | 0.19 ± 0.20 cm | 0.70 ± 1.09 cm | 0.99 ± 1.04 cm | 206.78 ± 253.21 g | 11.68 ± 16 days | NA |
| Correctly detected planes from cine-loops by FUVAI | 94 (76%) | 94 (76%) | 118 (83%) | 113 (78%) | NA | NA | NA |

Mean Absolute Error (MAE) for each predicted biometric value, EFW, and GA compared to clinicians and state-of-the-art model FUVAI showing superior correct detection rates and lower MAE with our approach, except for the BPD measurement. Source data are provided as a Source Data file.

method has the advantage of always giving the same output given the same cine-loop, which is not the case for human operators. This means that AI can reliably assess fetal growth status and potentially detect AFV abnormalities on fetal US cine-loops automating the third of the six items showcased in the ISUOG guidelines; and having the potential to address the shortage of sonographers in countries of the Global South.

HC, BPD, AC, and FL have been shown to be more reliable and reproducible amongst expert operators than SDP measurement with intra and inter CC > 0.990 amongst expert sonographers and clinically acceptable 95% limits of agreement[35,36]. Our models showed intra CC superior to 0.94 for all the biometry metrics (AC = 0.982, HC = 0.987, BPD = 0.975, FL = 0.945) and reached narrower 95% limits of agreement than those reported in studies assessing their reliability and reproducibility between human sonographers.

The models we developed were specifically designed to extract the best biometric planes according to the ISUOG criteria. Although other models have been developed to automate quality control of 2D fetal ultrasound images through anatomical structure recognition, our study is the first to explicitly use the ISUOG quality criteria specifically for biometry plane classification[37,38]. Such an approach, if integrated into the clinical workflow, could be used to automate the biometry plane's quality control. It could allow fast and inexpensive quality audits, accelerate the workflow of trained sonographers, and be a pedagogical tool to the sonographer in training. This could prove particularly useful in resource-stranded regions such as Africa, where only 38.3% of fetal US operators have received formal training, and only 40.4% of them have received a short theoretical course[3].

A similar study to ours compared the performances of a multi-task deep neural network (DNN) on FB assessment, testing it on 50 free-hand ultrasound videos with results comparable to those of trained sonographers. Our models outperformed the one described in the study (FUVAI)[14] when comparing the proximity of the results showcased by the model versus the sonographers expressed in MAE (Table 2) even if the DICE score coefficients and IoU were lower for the same tasks, potentially indicating greater generalizability of our models. FUVAI's choice of standard biometry planes didn't rely on the quality of the plane but rather on the confidence of the model when selecting it; in other words, on how closely it resembled images from the training set, which are not necessarily the best standard planes according to the ISUOG guidelines.

Another vast prospective study by Pokaprakarn et al.[18] took an original approach and assessed the performance of a DNN to estimate GA from blind loops taken by non-trained operators. The DNN proved to be more accurate than expert sonographers at estimating GA with an MAE of 3.9 ± 0.12 days vs 9.85 ± 14.36 days with our approach, which could be a game changer in resource-stranded environments. A recent study using the same dataset from the Fetal Age and Machine Learning Initiative (FAMLI) developed a DL tool to assess fetal biometry and fetal presentation on mobile devices showing non-inferiority to trained sonographers performing the measurements (−1.4 ± 4.5 days)[20]. These two studies manifest the potential of AI to be used in the clinical workflow to democratize access to quality fetal ultrasound worldwide. However, due to the nature of DNNs and the choice of blind sweeps, it is challenging to get a sense of how the model came up with its output, and impossible to extract AC or EFW for FGR risk assessment. Instead, our models mimic trained sonographers thanks to the separation of the FB workflow into classification, quality scoring, and segmentation tasks. They are thus understandable, with errors in the models' outputs easily detectable by sonographers.

SDP estimation has the widest variability with reported inter-observer limits of agreement of −51% to +52% and an ICC of 0.42[13,39]. Our approach for AFV assessment is vastly more reliable, with limits of agreement of only ±16.96 % and an ICC of 0.692 for SDP measurement. This high variability amongst human operators might be explained by

the "subjective" choice of the SDP. We brought the subjective choice closer to an objective one by segmenting and measuring every single AF pocket in a given cine-loop. In contrast, several studies present automated techniques to segment AF pockets and measure the pocket's depth. Cho et al.[33], for example, developed a CNN showcasing results similar to those of sonographers in segmenting AF pockets (DICE similarity coefficient: 0.877 ± 0.086) and with an MAE of 2.666 ± 2.986 cm in the measurement of the pocket's depth versus a DICE score of 0.783 in our study but an MAE of 1.46 ± 1.10 cm on prospectively acquired video loops. A second study by Sun et al.[40] complemented the work of Cho et al.[28] and developed a dual-path network approach to AF pocket segmentation to better account for reverberation artifacts in AF pockets, achieving a higher DICE similarity coefficient of 0.8599 ± 0.1074 on their dataset. However, both of these studies use 2D fixed images, only automating the segmentation part of the clinical workflow of AFV assessment. Ours proved to be clinically more precise and useful as they detected polyhydramnios with a sensitivity and specificity of 86.6%, and 85.7%, respectively.

As far as we are aware, our research is pioneering in its prospective evaluation of a model designed for estimating Amniotic Fluid Volume (AFV) using ultrasound videos. In practical application, healthcare professionals could verify the image chosen by the model, possibly rectifying any mistakes made either by themselves or the model, or they could browse through the selected pockets until they locate one that meets their satisfaction. Interestingly, this method could also be adapted for AFV evaluation from blind repeated craniocaudal perpendicular sweeps, thereby enabling even healthcare workers with minimal training to execute it.

A potential constraint of our methodology, stemming from the substantial size and computational demands of the models, is the requirement for an Internet connection for their deployment. This restricts their application to regions with Internet availability. Nonetheless, within the realm of telemedicine, Internet connectivity is typically a prerequisite, occasionally depending on a satellite link. Via telemedicine platforms, a proficient sonographer instructs an operator using a video stream of the ultrasound scan. Even under these circumstances, our automated technique can be employed to simplify the capture of ideal measurements and reduce the scan duration, thereby broadening access to quality imaging for prenatal care.

## Methods

In their comprehensive analysis of deep learning algorithms applied to fetal ultrasound-image examination, Fiorentino et al.[41] highlighted recent advancements in the use of these algorithms for identifying 2D fetal standard planes. Predominantly, the research focused on Convolutional Neural Networks (CNNs) for single-task operations, while some explored multi-task standard plane detection by identifying crucial anatomical structures. For instance, Baumgartner et al.[42] adapted the VGG16 architecture to detect 13 fetal standard planes, achieving a mean F1 score of 0.80 with their model, SonoNet. A few authors proposed multi-task models incorporating attention mechanisms for the detection of abdomen, head, and femur standard planes. Cai et al.[43], for example, trained a multi-task neural network with a temporal attention module and achieved F1 scores of 0.84, 0.89, and 0.81, respectively, on a test set of 280 videos lasting 3–7 s.

Fiorentino et al.[41] also reported on numerous research papers related to 2D fetal biometry estimation, with most focusing on a single structure, such as brain segmentation for HC measurement, abdomen segmentation for AC measurement, and femur segmentation for FL measurement. HC measurement has been the subject of intensive research due to the availability of the publicly annotated dataset HC18, despite the fact that the gold standard for GA and EFW formulae includes other biometric parameters such as AC and FL. Zeng et al.[15] proposed a modified version of V-Net that incorporates an attention mechanism and achieved a Dice score of 0.98 and an MAE of 1.95 mm.

Similarly, Moccia et al.[26] adapted a Mask-R-CNN method and achieved comparable results with a Dice score of 0.98 and an MAE of 1.95 mm. Notably, only one study reported in the review used videos for the prospective evaluation of models for BPD and HC measurements[44]. In contrast, very few papers have addressed abdomen and femur segmentation for AC and LF measurements, respectively. Using small datasets, Kim et al.[16] proposed an abdomen segmentation model that achieved a Dice score of 0.92, while Zhu et al.[45] proposed a femur segmentation model that achieved a Dice score of 0.92 and an MAE of 0.46 mm.

Most methods employed U-Net-based architectures, which are known for their semantic segmentation performance. However, Mask-R-CNN-based architectures were also utilized, as they allow for the segmentation of individual objects and the assessment of classification performance. Recently, some researchers have attempted a different approach, directly extrapolating measurements using regression models rather than running a segmentation model first and then approximating the measurement. To our knowledge, this approach has only been tested for HC estimation[46].

Very few papers have focused on multiple biometry estimation. To our knowledge, only one paper proposed a method for segmenting multiple anatomical structures for fetal biometry[14]. The authors proposed the FUVAI model, which combined U-Net with ConvLSTM architectures and was trained on a large private dataset to estimate HC, AC, and FL measurements using 274,275 2D ultrasound images. The model was tested on 57,001 2D images and achieved a Dice score of 0.96 and an MAE of 2.5 mm.

For 2D amniotic fluid volume (AFV) assessment, Fiorentino et al. reported only three research papers[28,40,47]. Using just 310 2D images, Cho et al. trained an adapted U-Net architecture called AF-Net and evaluated it on a test set of 125 2D images, achieving a Dice score of 0.87 and an MAE of 2.6 cm[28].

The use of different private datasets for evaluating these approaches, combined with the lack of public datasets, makes comparison challenging. All of the best performances were achieved using large, private datasets[42,43,48,49], indicating that a data-centric approach leads to better generalization. However, very few studies have focused on multi-organ analysis and end-to-end pipelines, with a lack of interpretation of results. Most are retrospective "in silico" studies conducted on Caucasian populations. To our knowledge, no research has proposed end-to-end pipelines for multiple fetal biometry structure segmentation, standard plan classification, and quality criteria assessments following ISUOG guidelines. Furthermore, our AFV assessment outperforms state-of-the-art results and is the first to be integrated and validated with a prospective study from cine-loops designed for minimally trained healthcare workers. Additionally, no prior prospective study has aimed to automate FB and AFV assessment using a large dataset of 2D ultrasound images of African patients examined in low-resource settings (see supplementary information file for details).

### Models and training

In the training of the seven Mask-R-CNN models, we adopted an image-centric training procedure[12]. Images were resized such that their scale (shorter edge) is 800 pixels[27]. PyTorch (version 1.10) framework was used for model training, validation, and testing. The models were trained with 80% of the data, validated with 10%, and tested with 10%. We trained on the NVIDIA Tesla K80 GPU for 2000 iterations, with a learning rate of 0.01 which was decreased by ten at the 500 iterations. We used a weight decay of 0.0001 and a momentum of 0.9. The used loss function is similar to the one described in Ref. 24. It combines the classification loss, the bounding-box loss, and the average binary cross-entropy loss of the mask.

In the training of the three classification models to infer the quality criteria of the abdomen and brain plans and to classify 11926

annotated images as containing AF pockets or not, the fully connected top layers were first replaced by an average pooling layer, then followed by a dense layer with a sigmoid activation function containing four outputs for the abdomen model, five outputs for the brain model and two outputs for the AF pocket model. TensorFlow (version 2.0) was used for model training, validation, and testing. We used 60% of the data for training, 20% for validation, and 20% for testing. The input images were resized to 224 × 224 pixels for the VGG16 model and to 299 × 299 for INCEPTIONV3 and RESNET50V2. For the brain and abdomen quality criteria classification, these images were first cropped based on their corresponding masks and then resized. The models were trained on the NVIDIA Tesla K80 GPU until convergence over 100 epochs using a batch size of 32 and an initial learning rate of $10^{-3}$ that is reduced by a factor of 0.2 once learning stagnates. The training is stopped early when there is no improvement in the validation loss for the last 15 epochs, or when we reach 100 epochs. We use binary cross-entropy loss and Adam as the optimizer of Keras (version 2.3.1). To prevent overfitting, we apply, on the fly, various data augmentation techniques using the following transformations: rotation between −15 and 15 degrees, zoom by 10%, brightness range between 0.2 and 0.8, as well as horizontal and vertical flipping.

From the training results of all these models, we adopted the R_50_DC5_3x model for the segmentation of the brain, abdomen, and femur, the INCEPTIONV3 models for the quality criteria classification and the AF pocket detection, and X_101_32x8d_FPN_3x model for the segmentation of AF pocket.

### Approximations of the biometric measurements

The biometric measurements are extracted from the output masks of the segmentation models. For the abdomen and brain, AC and HC are computed from the circumferences of the ellipses approximated by first, finding the contours and then direct least square fitting[42,43]. BPD is the minor axis of the brain ellipse. For the femur, FL represents the measure of the major axis of the extremities bounding box, and for the AF pocket, the SDP is the measure of the vertical axis of the bounding box. After the measurements are approximated in pixels, they are converted into centimeters using the DICOM's pixel spacing tag (see supplementary information file for details).

### Study design

We validated the DL models on prospectively consecutively acquired transabdominal US videos from pregnant patients (>18 years, evolutive pregnancy >14 weeks, non-emergency related scan indication,) gathered at four healthcare centers in Morocco (Casablanca and Oujda) from October 2021 to April 2022 by 7 different radiologists and obstetricians (experience in fetal US > 10 years) and annotated during the examination using the machine's ellipse and caliper facilities. The participating physicians were asked to measure HC, BPD, AC, and FL following the ISUOG criteria as well as the single deepest AF pocket (SDP) to assess AFV.

On top of their routine examination, the physicians had to take four additional cine-loops: three additional cine-loops containing all the standard biometry planes, and a cine-loop containing all AF pockets: an axial cephalic loop going from the base of the skull to the vertex, an axial abdominal loop going from the four-chamber view of the heart to a cross-section of the kidneys, a sagittal femur loop, and an amniotic loop sweeping perpendicularly through all the right, then the left AF pockets (Fig. 1).

The physicians had no knowledge of the predicted values for all biometric parameters until the end of the study, the team evaluating the models' performances was also tasked to gather the prospective data and hence had access to the predicted and measured values for each. The biometric parameters were inferred by the models at the end of the study, seven months after its beginning. On the modeling side, the best segmentation and classification models that were trained on retrospective data were run on each video to extract HC, BPD, AC, or FL measurements depending on the plane. All the detected AF pockets on the "amniotic" cine-loops were segmented and their depth was computed, retaining the deepest one as the predicted SDP. This approach is directly inspired by the standard steps taken by expert-trained sonographers to select the single deepest pocket. They consist of the following tasks: (1) Sweep through all AF pockets, (2) Subjectively select the SDP, and (3) Measure the SDP's depth. Oligohydramnios was defined as an SDP < 2 cm and polyhydramnios as an SDP > 8 cm[50].

The primary outcomes were the models performances in fetal biometry and single deepest pocket measurements, expressed in mean absolute error, limits of agreement with the sonographers, and Intraclass correlation coefficients.

The secondary outcomes were the performances of the models at detecting AFV and fetal growth abnormalities (FGR) using sensitivity, and specificity metrics.

This study follows the STARD 2015 guidelines[51] as detailed in the supplementary information STARD checklist.

Approval for this study was granted by the Institutional Review Board of Oujda's Faculty of Medicine (Comité d'Ethique pour la Recherche Biomédicale d'Oujda). Study participants provided their informed written consent and were not compensated for their participation to this study.

The full protocol of this study can be found on clinicaltrials.gov under the ID: NCT05059093.

### Evaluation and statistical analysis

DICE score coefficients and Intersection of Union (IoU) were computed for the Mask-RCNNs on the retrospective dataset. For the classification tasks, the receiving operating characteristics (ROC) curves were computed.

The intended sample size was estimated at 122 patients with all corresponding measurements and cine-loops correctly performed (see supplementary information file: Sample Size Estimation, for more details). We computed the mean absolute errors (MAE) between the models' measurements and the operators on the prospective cine-loops using the R package 'Metrics' (version 0.1.4) of R software (R version 4.2.1). Intraclass correlation coefficients (ICC) were calculated using the Package 'merTools' (version 0.5.2). ICC is a desirable measure of reliability that reflects both the degree of correlation and agreement between measurements. Wilcoxon rank sum test was calculated for each measurement using the 'PairedData' (version 1.1.1) R package. We also compared the performance of our approach to the FUVAI[14] model using the percentage of correctly classified planes and MAE using the R package 'Metrics' (version 0.1.4). Bland-Altman plots were used for the visual assessment of the models' reliability and the 95% limits of agreement were calculated and expressed in percentage using the 'blandr' package (version 0.5.1) of the R software. Firstly, The measurements from the operators and the model were passed to the blandr.statistics function to generate Bland-Altman statistics. After which, plots were generated using the package ggplot2 (version 3.3.6). Assessment of the models' performances was carried out alongside prospective data collection.

### Reporting summary

Further information on research design is available in the Nature Portfolio Reporting Summary linked to this article.

## Data availability

Source data are provided with this paper. Part of the de-identified fetal ultrasound data used in this study comes from a publicly available dataset on Zenodo published by Burgos-Artizzu, X. P. et al.[17] available

at https://zenodo.org/record/3904280. The rest of the de-identified fetal ultrasound data collected for the purpose of this study is available under restricted access due to privacy, ethical and legal considerations, access can be obtained by contacting the corresponding author at saadslimani@deepecho.io who will provide a response within 14 days and supply the data use agreement limiting its use to non-commercial research purposes.

## Code availability

The pre-trained models used in this study are available publicly: Classification models codes are available here: https://github.com/keras-team/keras/tree/v2.13.1/keras/applications. Segmentation models were developed using detectron2, available here: https://github.com/facebookresearch/detectron2/. The corresponding author will provide their finetuned weights of the models used in this study, for research and reproducibility purposes upon request at saad.slimani@deepecho.io within 14 days, subject to a data use agreement for non-commercial use.

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

## Acknowledgements

This study was entirely funded by Deepecho.inc. Authors would like to thank Leila Noureddine, Yasmine Guenni, Abed Tlemcani, Amine Fourari, Youssef Sakhi, and Nacer Abid for their hard and continuous work on image annotation. We would also like to thank Helena Pinheiro (hpinheiro.com) for the illustrations.

## Author contributions

S.S. conceived of the presented idea, original concept, was responsible for the study design, supervised the project, collected data, and co-authored the manuscript. S.H. was the lead programmer, performed computational analysis, designed models, extracted the data, created figures, and co-authored the manuscript. A.M. supervised the project from data collection to models training and evaluation and co-authored the manuscript. T.R. contributed to programming, model design, and data extraction. These authors contributed equally: D.L., H.S., A.B., A.L., M.J., S.B., M.A.: performed ultrasound examinations, collected data, and helped supervise the project. Y.B.: original concept, supervised the project. B.B. advised the project, performed statistical analysis, helped develop the statistical analysis plan, and contributed to manuscript writing. A.R.: advised the project. M.M.: contributed to manuscript writing and advised the project. E.B. helped supervise the project and verified manuscript and analytical methods.

## Competing interests

The S.S., A.M., Y.B., and E.B. are shareholders and employees of Deepecho inc. S.H. and T.R. are employees of Deepecho.inc. The remaining authors declare no competing interests.

## Additional information

[1]Deepecho, 10106 Rabat, Morocco. [2]Ibn Rochd University Hospital, Hassan II University, 20100 Casablanca, Morocco. [3]Telecommunications Systems Services and Networks lab (STRS Lab), INPT, 10112 Rabat, Morocco. [4]Ecole Normale Supérieure, LIMIARF, Mohammed V University in Rabat, 4014 Rabat, Morocco.

[5]Mohammed VI University Hospital, 60049 Oujda, Morocco. [6]Université Mohammed VI des Sciences de la Santé, Hôpital Universitaire Cheikh Khalifa, 82403 Casablanca, Morocco. [7]Abou Madi Radiology Clinic, 20060 Casablanca, Morocco. [8]Laboratory of Biodiversity, Ecology, and Genome, Department of Biology, Faculty of Sciences, Mohammed V University in Rabat, 1014 Rabat, Morocco. [9]African Sustainable Agriculture Research Institute (ASARI), Mohammed VI Polytechnic University (UM6P), 43150 Laâyoune, Morocco. [10]Radboud Institute for Molecular Life Sciences, Epigenomics & Single Cell Biophysics, 6525 XZ Nijmegen, the Netherlands. ✉e-mail: saad.slimani@deepecho.io

