## [Peer Review File · Nature Communications]

Fetal biometry and amniotic fluid volume assessment
end-to-end automation using Deep LearningEditorial Note: This manuscript has been previously reviewed at another journal that is not operating a transparent peer review scheme. This document only contains reviewer comments and rebuttal letters for versions considered at *Nature Communications*.

REVIEWER COMMENTS

Reviewer #1 (Remarks to the Author):

I am convinced by the rebuttal. Thanks for the effort to revise the paper and answer my questions.

Reviewer #2 (Remarks to the Author):

The Authors have satisfactorily responded to Reviewers 1 and 3 critiques. The manuscript was appropriately updated based on Reviewer comments.

I am unable to comment on some of the details around the “fine tuning” of deep learning models in their response to Reviewer 2.

With regards to my most important comments:

Figure 2. Ultrasound images appear too “dark” for the readers on my computer screen. Consider adjusting brightness and contrast of the images.

Figure 8. If I interpreted the Bland-Altman plot correctly, it seems to indicate -19% measurement bias for Single Deepest Pocket (cm) with wide 95% Limits of Agreement ranging between -71 to +37%. If true, the Authors should comment on the variable reproducibility of this parameter and its practical application in a clinical environment.

Figure 10. The FL dashed line is supposed to measure the femur diaphysis. The example diagram appears to overestimate the diaphyseal shaft measurement. Consider shortening horizontal measuring line in this diagram if standard biometry methods were indeed employed.

Reviewer #3 (Remarks to the Author):

The manuscript titled "End-to-End Automation of Fetal Biometry and Amniotic Fluid Volume Assessment Using Deep Learning" introduces a Deep Learning approach for automatically measuring fetal biometries and assessing fluid volume. While the manuscript is intriguing, it would greatly benefit from improved organization by the authors. Additionally, a crucial aspect missing from the paper is a comparison with the current state of the art of the field. Furthermore, the proposed methodology lacks significant innovation. As the authors highlight the inclusion of African patients as a contribution, it would be highly valuable for them to make the dataset publicly available. English sometimes is really poor.

INTRODUCTION:

The authors have overlooked the state-of-the-art in the field, and it is recommended that they familiarize themselves with the most recent survey titled "A review on deep-learning algorithms for fetal ultrasound-image analysis" by Fiorentino et al. This will provide them with valuable insights.

The manuscript lacks clarity in explaining the sentence, "Furthermore, no previous work has tried to develop an understandable approach to the automation of such tasks, previous attempts using 'blind' cine-loops do seem to allow for the democratization of gestational age estimation and their use by minimally trained sonographers." It is essential for the authors to elaborate on their intended meaning in this section. They should specify their intent, which is standard plane detection and biometry parameter estimation, and explain the contribution of their proposed approach. Furthermore, it would be helpful for the authors to clarify why they decided to combine these two tasks.

Why is the "Results" section placed before the "Method" section? It would be more appropriate to move the data description to the "Method" section

RESULTS:

Please include a dedicated subparagraph titled "Methods" where you provide a description of the architecture used and the dataset information. The "Results" section should focus solely on presenting the results.

Could you specify the number of women who participated in the study? Additionally, it would be helpful to provide information on the gestational weeks covered in the study.

Regarding the statement "we used a set of Mask-RCNN architectures," further clarification is needed to better understand its meaning. It is important to note that Mask-RCNN is a commonly employed architecture in several state-of-the-art articles, including the works of Al-Bander et al. (2019) and Moccia et al. (2021). In order to provide a comprehensive understanding of the research methodology, it is crucial for the authors to specify the specific advancements, modifications, or contributions they have made in comparison to these prior studies. This will help establish the unique value and novel aspects of their own research in relation to the existing literature. Why did authors chose this specific architecture over others available?

Could you please clarify whether the images that contain two or more instances of an object, such as two femurs or two heads in the case of twins, are included in both the training and test sets? It would be helpful to specify their inclusion.

Additionally, the authors mentioned that "classification models for quality criteria detection were assessed on the test set of the retrospective data" for each biometry plane. Could you explain why the assessment was conducted only on the test set? It would be beneficial to provide a rationale for this approach.

The manuscript lacks a comparison with the state-of-the-art approaches, making it challenging to assess the proposed contributions in relation to existing methods. Although the authors claim to have employed an end-to-end biometry architecture estimation, it is important to note that they relied on post-processing methods for object contour detection. It is worth mentioning that several approaches currently exist that directly extrapolate HC measures without the need for post-processing methods.

Why did author decide to compare only InceptionV3, VGG16 and Resnet50? As for now, many architectures have been used to solve scane plane detection (e.g. SonoNet etc).

METHODS:

MODELS AND TRAINING:

It is unclear why the authors chose to use binary cross entropy as the loss function, considering that the problem involves multiple quality criteria, making it a potential multiclass problem. For instance, in the study, four quality criteria were assessed (kidneys not visible (A_KN), plane showing portal sinus (A_PS),

plane showing stomach bubble (A_SB), and symmetrical plane (A_SYM)). It would be beneficial for the authors to explain the rationale behind using binary cross entropy in this context and consider an appropriate loss function for multiclass classification. How was the augmentation performed (e.g. on the fly / offline)?

Additionally, it would be helpful if the authors specified the loss function used to train the Mask-RCNN model. Why did author decide to work with 2D images as opposed to videos?

Evaluation and statistical analysis:

In order to facilitate a fair comparison of the proposed methodology with state-of-the-art approaches, it is recommended that the authors include a reference to the HC18 challenge metrics for evaluating the metrics. This would provide a standardized framework for evaluating the performance of the proposed methodology.

Minor comments:

Please provide better captions of figures.

Figure 1: What do “doctors, semi-automatic, total” mean? Is the classification performed only for brain and abdomen? Why?

REBUTTAL LETTER

All changes in the revised manuscript have been highlighted in yellow.

Reviewer #1 (Remarks to the Author)

I am convinced by the rebuttal. Thanks for the effort to revise the paper and answer my questions.

Thank you for taking the time to review our paper and for your positive feedback on our revisions. We appreciate your thoughtful comments and are glad our rebuttal addressed your concerns.

Reviewer #2 (Remarks to the Author)

The Authors have satisfactorily responded to Reviewers 1 and 3 critiques. The manuscript was appropriately updated based on Reviewer comments.

I am unable to comment on some of the details around the “fine tuning” of deep learning models in their response to Reviewer 2.

Thank you

With regards to my most important comments:

Figure 2. Ultrasound images appear too “dark” for the readers on my computer screen. Consider adjusting the brightness and contrast of the images.

Thank you for your comment; we have adjusted the contrast of the images you mentioned directly on the manuscript (see pages 9 and 21) and hope they look clearer now:

(1) Original image

(2) ROI detection

(3) Calipers and dots detection

(4) Cleaned image

(5) Mask extraction (validated by annotator)

Brain Standard Plane
BPD: 9,40cm
HC: 33,65cm

(6) Standard plan and biometric measurement recognition using OCR (validated by annotator)

Figure 2: Preprocessing steps for the generation of ground truth masks and standard plan and biometric measurement recognition. (1) original image of a brain plan with pixelated yellow calipers, dotted circumference, and biometric measurements (2) region of interest detection (3) Calipers and dots detection within the region of interest (4) cleaned image using inpainting techniques (5) mask extraction and validation by the annotators and (6) Standard plan and biometric measurement recognition using OCR techniques.

Figure 8. If I interpreted the Bland-Altman plot correctly, it seems to indicate -19% measurement bias for Single Deepest Pocket (cm) with wide 95% Limits of Agreement ranging between -71 to +37%. If true, the Authors should comment on the variable reproducibility of this parameter and its practical application in a clinical environment.

Thank you for your insightful remark. The numbers you cited of -71 to +37% represent extreme cases with a maximum difference between the model's SDP and the physicians'. However, the 95% limits of agreement calculated for our approach were notably tighter, at $\pm 16.96\%$, in contrast with the -51% to +52% observed for expert human operators. This signifies that our method is more reliable and offers greater reproducibility, making it a robust tool in a clinical environment.

The essence of measuring the SDP lies in its capacity to detect conditions such as oligohydramnios and polyhydramnios. While our study did not encounter cases of oligohydramnios, our methodology succeeded in diagnosing polyhydramnios with 86.6% sensitivity and 85.7% specificity compared to experts' assessments.

Moreover, we're proud to point out that our study is the first, to our knowledge, that prospectively evaluates a deep-learning-based approach for SDP assessment and polyhydramnios detection from ultrasound videos. See page 24 for more details:

"SDP estimation has the widest variability with reported inter-observer limits of agreements of -51% to + 52% and an ICC of 0.4213,38. Our approach for AFV assessment is vastly more reliable, with limits of agreement of only $\pm 16.96\%$ and an ICC of 0.692 for SDP measurement. This high variability amongst human operators might be explained by the "subjective" choice of the SDP. We brought the subjective choice closer to an objective one by segmenting and measuring every single AF pocket in a given cine-loop." This progression in the field holds considerable promise for improving the reliability of diagnoses and potentially patient outcomes.

While the Bland-Altman plot does indicate a level of measurement bias and variability, our approach's overall reliability, reproducibility, and unique deep-learning implementation make it a practical and innovative tool in a clinical setting.

Figure 10. The FL dashed line is supposed to measure the femur diaphysis. The example diagram appears to overestimate the diaphyseal shaft measurement. Consider shortening

horizontal measuring line in this diagram if standard biometry methods were indeed employed.

Thank you for the great comment. The point we are trying to make is that the relatively larger difference between the doctors' and models' measurements in the case of the femur can be explained by the fact that our models actually stick to the ISUOG guidelines in terms of quality criteria which is not systematically the case with the physicians. We noticed that in extreme cases, where the difference between the two measurements was larger than 1 cm, our approach objectively came up with femoral planes and caliper placement of better quality than physicians. Thanks to your comment, we decided to change the example showcased in the previous manuscript and chose a set of images that better emphasizes our point. See below:

Expert operator choice of plane and caliper placement

Our approach output on the same patient on a video loop

As you can see, our approach came up with a strictly horizontal femur and did not encompass the trochanter in the femur measurement, thus abiding by measurement guidelines. The doctor, however, chose a more tilted femur and included part of the trochanter in their measurement. We can then assume that the larger difference between the models' and physicians' measurements observed in the case of the femur can be explained, partly by the fact that doctors did not always choose the best planes while our approach was closer to the ideal quality criteria described in the ISUOG guidelines.

Reviewer #3 (Remarks to the Author)

The manuscript titled "End-to-End Automation of Fetal Biometry and Amniotic Fluid Volume Assessment Using Deep Learning" introduces a Deep Learning approach for automatically measuring fetal biometry and assessing fluid volume. While the manuscript is intriguing, it would greatly benefit from improved organization by the authors.

Additionally, a crucial aspect missing from the paper is a comparison with the current state of the art of the field.

Furthermore, the proposed methodology lacks significant innovation.

Respectfully we do not claim to have developed a new deep-learning model or have made significant modifications to existing models. However, we emphasize the clinical innovation that comes from the utilization of existing models and their inclusion in the clinical workflow. Our approach is thus data-centric rather than model-centric.

We firmly believe that concrete advances in the utilization of DL in clinical practice will come from rigorous clinical evaluation of the models as well as their usability: prospective studies are the gold standard to assess the performance of a new measurement tool in the clinical workflow.

That said, the manuscript details several other advancements in the state of the art beyond this. To the best of our knowledge, no prior study has developed a new way to integrate machine learning models in the amniotic fluid volume assessment workflow from image detection to segmentation to the measurement of the SDP on a video loop,, a key improvement over the state of the art that we present here. The “amniotic loop” for the measurement of the single deepest pocket was explicitly designed as a “blind’ loop that any minimally trained healthcare worker can perform: two longitudinal sweeps, perpendicular to the ground, on each side of the pregnant person’s abdomen. Our biometry model is the first of its kind to use the ISUOGI guidelines to assess the quality of all three standard biometry planes. The purpose of this approach is explicitly to democratize access to this expertise in the choice of biometry plane.

To compare with the state-of-the-art models and highlight our main contributions, we added a subsection, “Reference Methods,” under the “Methods” section and a table 3 that summarizes the best-performing deep learning models reported in the state-of-the-art literature for the classification and segmentation tasks to detect 2D standard plans, and assess FB and AFV.

As the authors highlight the inclusion of African patients as a contribution, it would be highly valuable for them to make the dataset publicly available.

Thank you for your remark, as stated in the data-sharing agreement, we are making the study data available for fellow researchers upon request for non-commercial use.

English sometimes is really poor.

We apologize if the quality of the writing did not meet your expectations. We hired a native English speaker and professional writer to help us rewrite the manuscript, and we hope you will now find it to your liking.

INTRODUCTION

The authors have overlooked the state-of-the-art in the field, and it is recommended that they familiarize themselves with the most recent survey titled "A review on deep-learning algorithms for fetal ultrasound-image analysis" by Fiorentino et al. This will provide them with valuable insights.

Thank you for bringing this exciting survey to our attention. First, we would like to emphasize again that our work is centered around the clinical use of the models rather than single improvements on subtasks like standard plane detection; that is why we developed one unified approach for the automation of fundamental yet complex fetal ultrasound tasks rather than minor incremental improvements to subtasks of the fetal biometry or amniotic fluid volume assessment workflow.

After careful consideration of the studies described in the review, we note that only one study in biometry uses videos for the prospective evaluation of the models, the study by Rashedd et al. ¹, but it only tackles the measurement of the BPD and HC. Our study develops an approach to automate the measurement of BPD, HC, FL, AC, and SDP, which no other study described in the survey has achieved.

We also note that the survey published in January 2022 does not mention the most recent study by Plotka et al. ² published in February 2022, on fetal biometry parameters extraction from ultrasound videos which constitutes, in our opinion, the state-of-the-art in the field of fetal biometry automation using deep learning. This study which is the closest to ours, is extensively described in our work and directly compared to our approach in Table 2.

	HC	BPD	FL	CA	EFW	GA	SDP
MAE ± standard dev with our approach	0.67 ± 0.69 cm	0.33 ± 0.22 cm	0.27 ± 0.40 cm	0.91 ± 0.81 cm	147.18 ± 177.97 g	9.85 ± 14.36 days	1.46 ± 1.10 cm
Correctly detected planes from cine-loops with our approach	123 (100%)	123 (100%)	142 (100%)	144 (100%)	NA	NA	90 (100%)
MAE ± standard dev FUVAI	0.70 ± 0.67 cm	0.19 ± 0.20 cm	0.70 ± 1.09 cm	0.99 ± 1.04 cm	206.78 ± 253.21 g	11.68 ± 16 days	NA
Correctly detected planes from cine-loops by FUVAI	94 (76%)	94 (76%)	118 (83%)	113 (78%)	NA	NA	NA

Table 2: Mean Absolute Error (MAE) for each predicted biometric value, EFW, and GA compared to clinicians and state-of-the-art model FUVAI showing superior correct detection rates and lower MAE with our approach, except for the BPD measurement.

After careful evaluation of existing studies, we still believe that the state-of-the-art in the field of biometry assessment using deep learning is represented by the FUVAI model because of the following:

- It encompasses all four biometry parameters: HC, BPD, FL, and AC, which is not the case in the majority of the studies described in the survey

- FUIVAI is evaluated on prospectively acquired videos.
- The performances of the model are directly compared and are equivalent to those of trained sonographers.

We also failed to find a study in the review that combines the following elements: ultrasound videos are used for inference, that better reflects real situation utilization, classification of standard biometry planes and AF pockets, specifically refers to the ISUOG criteria for all three standard biometry planes, combines the segmentation of said elements to extract biometry and SDP, and is tested prospectively on a large number of videos.

We refer again to the section “Reference Methods” and Table 3 for a summary of the state-of-the-art methods and our main contribution, which you can find below and in pages 26 to 29 of the revised manuscript:

Reference Methods

In their recent review of deep learning algorithms for fetal ultrasound-image analysis, Fiorentino et al.⁴¹ reported on the latest research utilizing deep learning algorithms for the detection of 2D fetal standard planes. The majority of the studied architectures were CNN-based and focused on a single task, while others proposed multi-task standard plane detection through the identification of key anatomical structures. For instance, Baumgartner et al.⁴² adapted the VGG16 architecture to detect 13 fetal standard planes, achieving a mean F1 score of 0.80 with their model, SonoNet. A few authors proposed multi-task models incorporating attention mechanisms for the detection of abdomen, head, and femur standard planes. Cai et al.⁴³, for example, trained a multi-task neural network with a temporal attention module and achieved F1 scores of 0.84, 0.89, and 0.81, respectively, on a test set of 280 videos lasting 3-7 seconds.

Fiorentino et al.⁴¹ also reported on numerous research papers related to 2D fetal biometry estimation, with most focusing on a single structure, such as brain segmentation for HC measurement, abdomen segmentation for AC measurement, and femur segmentation for FL measurement. HC measurement has been the subject of intensive research due to the availability of the publicly annotated dataset HC18, although the gold standard for GA and EFW formulae includes other biometric parameters such as AC and FL. Zeng et al.¹⁵ proposed a modified version of V-Net that incorporates an attention mechanism and achieved a Dice score of 0.98 and an MAE of 1.95mm. Similarly, Moccia et al.²⁶ adapted a Mask-RCNN method and achieved comparable results with a Dice score of 0.98 and an MAE of 1.95mm. Notably, only one study reported in the review used videos for the prospective evaluation of models for BPD and HC measurement⁴⁴. In contrast, very few papers have addressed abdomen and femur segmentation for AC and LF measurement, respectively. Using small datasets, Kim et al.¹⁶ proposed an

abdomen segmentation model that achieved a Dice score of 0.92, while Zhu et al.⁴⁵ proposed a femur segmentation model that achieved a Dice score of 0.92 and an MAE of 0.46mm.

Most methods employed U-Net-based architectures, which are known for their semantic segmentation performance. However, Mask R-CNN-based architectures were also utilized, as they allow for the segmentation of individual objects along with the assessment of classification performance. Recently, some researchers have attempted a different approach, directly extrapolating measurements using regression models rather than running a segmentation model first and then approximating the measurement. To our knowledge, this approach has only been tested for HC estimation⁴⁶.

Very few papers have focused on multiple biometry estimation. To our knowledge, only one paper proposed a method for segmenting multiple anatomical structures for fetal biometry¹⁴. The authors proposed the FUVAI model, which combines U-Net with ConvLSTM architectures and was trained on a large private dataset to estimate HC, AC, and FL measurements using 274,275 2D ultrasound images. The model was tested on 57,001 2D images and achieved a Dice score of 0.96 and an MAE of 2.5mm.

For 2D amniotic fluid volume (AFV) assessment, Fiorentino et al. reported only three research papers^{28,40,47}. Using just 310 2D images, Cho et al. trained an adapted U-Net architecture called AF-Net and evaluated it on a test set of 125 2D images, achieving a Dice score of 0.87 and an MAE of 2.6cm²⁸.

Table 3 summarizes the best-performing deep learning models reported in state-of-the-art literature for classification and segmentation tasks related to the detection of 2D standard planes and the assessment of fetal biometry and amniotic fluid volume. The use of different private datasets for evaluating these approaches, combined with the lack of public datasets, makes comparison challenging. All of the best performances were achieved using large, private datasets^{42,43,48,49}, indicating that a data-centric approach leads to better generalization. However, very few studies have focused on multi-organ analysis and end-to-end pipelines, with a lack of interpretation of results. Most are retrospective “in silico” studies conducted on Caucasian populations. To our knowledge, no research has proposed end-to-end pipelines for multiple fetal biometry structure segmentation, standard plan classification, and quality criteria assessments following ISUOG guidelines. Furthermore, our AFV assessment outperforms state-of-the-art results and is the first to be integrated and validated with a prospective study from cine loops designed for minimally trained healthcare workers. Additionally, no prior prospective study has aimed to automate FB and AFV assessment using a large dataset of 2D ultrasound images of African patients examined in low-resource settings.

Reference	Task	Data	Measurement	Method	Performance	Main Contribution
Baumgartner et al. [42] (2017)	2D Standard Plan Detection	140,827 2D images and 2,438 videos	13 Standard Planes	SonoNet (Adapted VGG16)	Mean F1 = 0.80	Detection of 13 fetal standard planes.
Cai et al. [43] (2020)	2D Standard Plan Detection	280 2D videos (3-7)s	Abdomen, Femur and Brain standard planes	biCLSTM with Temporal Attention	Mean F1 = 0.85	Better performance in the classification of three standard planes
Zeng et al. [15] (2021)	2D Fetal biometry	999 2D images (HC18)	HC	V-Net with attention	Dice = 0.98 MAE = 1.77mm	Best segmentation performance of HC on a public dataset.
Moccia et al. [26] (2021)	2D Fetal Biometry	999 2D images (HC18)	HC	Adapted Mask-RCNN	Dice = 0.98 MAE = 1.95mm	Best segmentation performance of HC on a public dataset.
Plotka et al. [14] (2022)	2D Fetal Biometry	274,275 2D images from videos	HC, AC, FL, GA, EFW	FUVAI (U-Net + ConvLSTM)	Dice = 0.96 MAE = 2.5mm GA MAE = 0.35d EFW MAE = 25g	Best segmentation performance of FB, validated on a large dataset.
Pokaparakarn et al. [18] (2022)	2D Fetal Biometry	147,855 2D images (FAMLI)	GA	Deep learning model	GA MAE = 3.9d	end-to-end GA estimation.
Gomes et al. [20] (2022)	2D Fetal Biometry	147,855 2D images (FAMLI)	GA	Deep learning model	GA MAE = -1.4d	end-to-end GA estimation in a Mobile setting.
Cho et al. [28] (2021)	2D Amniotic Fluid	310 2D images	SDP	AF-Net (Adapted U-Net)	Dice = 0.87 SDP MAE = 2.66cm	Best segmentation performance, yet validated on a very small dataset.
Our Method	2D Standard Plan Detection, 2D Fetal Biometry and Quality Criteria	30,249 2D images	HC, BPD, AC, FL, GA, EFW	Mask-RCNN INCEPTIONV3	Mean F1 = 0.73 Dice = 0.89 MAE = 6.1mm GA MAE = 9.85d EFW MAE = 147.18g	use of ISUOG guidelines to assess the quality of all three standard biometry planes validated prospectively on a larger dataset of African patients.
Our Method	2D Amniotic Fluid	11,926 2D images	SDP	Mask-RCNN INCEPTIONV3	F1 = 0.81 Dice = 0.78 SDP MAE = 1.46cm	Blind loops to measure the SDP that is validated prospectively on a larger dataset of African patients.

Table 3: Best-performing deep learning models reported in the state-of-the-art literature for the classification and segmentation tasks to detect 2D standard plans, and assess fetal biometry or amniotic fluid volume. The two last rows present the results of our models and the last column reports the main contributions.

The manuscript lacks clarity in explaining the sentence, "Furthermore, no previous work has tried to develop an understandable approach to the automation of such tasks, previous

attempts using 'blind' cine-loops do seem to allow for the democratization of gestational age estimation and their use by minimally trained sonographers." It is essential for the authors to elaborate on their intended meaning in this section. They should specify their intent, which is standard plane detection and biometry parameter estimation, and explain the contribution of their proposed approach. Furthermore, it would be helpful for the authors to clarify why they decided to combine these two tasks.

Thank you for your insightful comment. Allow us to clarify our purpose. Our approach is a clinical and pragmatic one rather than an incremental and purely technical one. As a research team, we aim to develop tools that are of value to clinicians and clinically tested. Thus we are attentive to new approaches that tackle the problem of the need for qualified sonographers in the Global South. The article by Pokaparakarn et al.⁷, as well as other later articles⁸, presents such a new approach. We were impressed by the reach of these papers describing the use of "blind sweeps" to automate fetal biometry and gestational age estimation in low-resource settings, as they were published while we were writing our manuscript. In our opinion, they represent a paradigm shift, instead of the deconstruction and automation of the sonographer workflow, they present with a way to extract 'hidden' biomarkers of gestational age in a video loop:

"As we have verified through manual review, it is rare for a blind sweep to contain the ideal image planes necessary for standard fetal biometry. Although the nature of the deep-learning algorithm is such that we do not know exactly which image features the model uses to make its predictions, it seems likely to be incorporating many facets of the available data to accomplish its task, rather than simply mimicking that which is acquired when a trained sonographer performs biometry."

Our perspective is that of clinicians who will ultimately use these models, for whom the outcome and clinical relevance are the most relevant characteristics.

Our aim is to develop an understandable approach to fetal biometry and amniotic fluid volume workflows automation, hence: the detection of standard biometry planes and segmenting of the corresponding anatomical landmarks , selecting the best standard plane following the ISUOG recommendation, measuring the largest AF pocket, and computing the AC, FL, HC, BPD, SDP, GA and EFW.

We want to emphasize the difference between our understandable and precise approach with the "blind loop."

By incorporating the ISUOG criteria, we are taking an extra step that no one has done before for all three standard planes, thereby empowering sonographers and aiding in their training. The combination of segmentation and quality criteria allows us to mimic what an ideal physician would do under ideal conditions. By automating this expertise, we are automating gold standard practices in areas where there may be a shortage of trained personnel.

In light of your comment, we changed the sentence:

" Furthermore, no previous work has tried to develop an understandable approach to the automation of such tasks, previous attempts using 'blind' cine-loops do seem to allow for the democratization of gestational age estimation and their use by minimally trained sonographers."

To:

"Furthermore, no previous work has tried to develop an understandable approach to the automation of **both tasks respecting the quality guidelines set forth by the International Society of Ultrasound in Obstetrics and Gynecology (ISUOG) for the FB workflow or allowing an end-to-end automation of the AFV assessment workflow. Novel approaches** using "blind" cine-loops do seem to allow for the democratization of gestational age estimation and their use by minimally trained sonographers but fail to promote the autonomy and education of the operators

Why is the "Results" section placed before the "Method" section? It would be more appropriate to move the data description to the "Method" section

Respectfully, we followed the formatting described in the guide for authors in Nature Communications that specifically demands the results section be placed before the methods section: *"The main text of an Article should begin with a section headed Introduction of referenced text that expands on the background of the work (some overlap with the abstract is acceptable), followed by sections headed Results, Discussion (if appropriate) and Methods (if appropriate). The Results and Methods sections should be divided by topical subheadings; the Discussion should be succinct and may not contain subheadings. Methods are typically less than 3000 words. Figure legends are limited to 350 words each. As a guide, references should not exceed 70. Footnotes are not used."*

See: <https://www.nature.com/ncomms/submit/article>

RESULTS

Please include a dedicated subparagraph titled "Methods," where you provide a description of the architecture used and the dataset information. The "Results" section should focus solely on presenting the results.

Respectfully the Methods sections can be found after Discussion as per the Nature Communication guidelines

Could you specify the number of women who participated in the study? Additionally, it would be helpful to provide information on the gestational weeks covered in the study.

Thank you for your comment. The number of participants is 172, as described on pages 16 and 17, characteristics of the study population, such as age, gestational age, and comorbidities, are also described:

“Overall, the mean GA estimated by the operators was of 30 weeks and 3.13 days \pm 6 weeks and 3.1 days (range: 15 weeks and 2 days – 41 weeks and 2 days), the mean measured HC, BPD, AC, FL, EFW, and SDP were respectively of 26.37 ± 5.88 cm (range: 11.29 – 34.71 cm), 7.41 ± 1.72 cm (range: 3.09 – 10.07 cm), 23.98 ± 6.58 cm (range: 8.95 – 38.18 cm), 5.28 ± 1.44 cm (range: 1.52 – 7.86 cm), 1606.78 ± 957.56 g (range: 108.81 – 3783.86 g and 5.25 ± 2.22 cm (range: 2.15 – 17.37 cm).”

“From October 2021 to April 2022, 172 patients with singleton pregnancies were included in our prospective study, the average age of the participants was of 30.38 years (minimum: 18, maximum: 44, standard deviation: 6.05), most of the included patients did not have any comorbidity (87%), ten patients lived with diabetes mellitus (6%) and three lived with a hypertensive disorder (2%). 34 patients (20%) were nulliparous, 47 (27%) were primiparous, and 91 (53%) were multiparous.”

Figure 6: Study Flow Chart

Regarding the statement "we used a set of Mask-RCNN architectures," further clarification is needed to understand its meaning better. It is important to note that Mask-RCNN is a commonly employed architecture in several state-of-the-art articles, including the works of Al-Bander et al. (2019) and Moccia et al. (2021). In order to provide a comprehensive understanding of the research methodology, it is crucial for the authors to specify the specific advancements, modifications, or contributions they have made in comparison to these prior studies. This will help establish the unique value and novel aspects of their own research in relation to the existing literature. Why did the authors choose this specific architecture over others available?

Thank you for your comment. When we stated that we used a set of Mask-RCNN architectures, we were referring to the Mask-RCNN architecture described on pages 12 and 15 for the segmentation of fetal biometry (FB) and amniotic fluid volume (AFV), respectively. Specifically, we used the R_101_C4_3x, R_101_DC5_3x, R_50_C4_3x, and R_50_DC5_3x architectures for FB segmentation, and the R_101_C4_3x, R_101_DC5_3x, R_101_FPN_3x, R_50_C4_3x, R_50_DC5_3x, R_50_FPN_3x, and X_101_32x8d_FPN_3x architectures for AFV assessment.

While Mask-RCNN models are widely used in state-of-the-art literature, the datasets used for training and testing these models are often insufficient for efficient performance evaluation. For instance, Al-Bander et al⁹ used only 1334 2D ultrasound images from the HC18 dataset for fetal head assessment. In contrast, our Mask-RCNN models were trained on a dataset containing 30249 annotated images for brain, abdomen, and femur segmentation and 3773 images for AF pocket segmentation. We adopted a data-centric rather than a model-centric approach and did not claim to have made significant modifications to existing architectures. Some authors, such as Moccia et al.¹⁰, made changes to the main architecture of Mask-RCNN to solve the problem of fetal head delineation and achieved remarkable results. However, their model was evaluated on the small HC18 dataset and did not cover other fetal structures. Our dataset allowed us to train vanilla Mask-RCNN segmentation models for all fetal structures with varying shapes and locations in the image, facilitating the straightforward deployment of such a system. We emphasized these claims on page 12.

Could you please clarify whether the images that contain two or more instances of an object, such as two femurs or two heads in the case of twins, are included in both the training and test sets? It would be helpful to specify their inclusion.

Thank you for your comment. While multiple pregnancies were not an exclusion criterion for our study, no multiple pregnancies were recruited during the study period. We recognize the

possibility of encountering two femurs from the same fetus within a single image. In such cases, our trained Mask-RCNN model is capable of segmenting both femurs and for the purposes of our study, we considered the femur object with the highest confidence score.

The following figure illustrates a representative image selected from a video in our dataset, shown on the left. This image captured a single femur and was chosen based on specific criteria to ensure its suitability for segmentation evaluation. The image on the right depicts two femurs that were successfully detected and segmented by our model

Additionally, the authors mentioned that "classification models for quality criteria detection were assessed on the test set of the retrospective data" for each biometry plane. Could you explain why the assessment was conducted only on the test set? It would be beneficial to provide a rationale for this approach.

Thank you for your comment. As detailed in the Models Training subsection, our classification models for quality criteria were tested on a randomly sampled set from the retrospective data: "The models were trained with 80% of the data, validated with 10%, and tested with 10%." The test set was not seen during training. The metric used to assess model performance on the prospective dataset was the difference between predicted and measured biometric parameters. While we agree that an external assessment of images selected by both the models and expert operators would be a valuable addition to our work, we assumed that trained operators with over 10 years of experience in fetal ultrasound represent the gold standard in their selection of standard planes, as they were explicitly instructed to select planes according to the ISUOG quality criteria.

The manuscript lacks a comparison with the state-of-the-art approaches, making it challenging to assess the proposed contributions in relation to existing methods. Although

the authors claim to have employed an end-to-end biometry architecture estimation, it is important to note that they relied on post-processing methods for object contour detection. It is worth mentioning that several approaches currently exist that directly extrapolate HC measures without the need for post-processing methods.

Thank you for your comments. As we previously mentioned, we have added a Reference Methods section and Table 3 to summarize state-of-the-art methods and our main contributions. By end-to-end, we are referring to the perspective of the clinician, which is of utmost importance when attempting to bring technology from the bench to the bedside.

With regard to post-processing methods for approximating fetal biometry contours, two main approaches have been reported in the literature: running a segmentation model first and then approximating the measurement, or directly extrapolating the measurement using regression models. The first approach is more commonly used and covers nearly all fetal biometry measurements, while the second approach has been used more recently and exclusively for HC estimation⁶. We have added a paragraph to the Reference Methods subsection stating this fact: “Most methods employed U-Net-based architectures, which are known for their semantic segmentation performance. However, Mask R-CNN-based architectures were also utilized, as they allow for the segmentation of individual objects along with the assessment of classification performance. Recently, some researchers have attempted a different approach, directly extrapolating measurements using regression models rather than running a segmentation model first and then approximating the measurement. To our knowledge, this approach has only been tested for HC estimation.

Why did the author decide to compare only InceptionV3, VGG16, and Resnet50? As for now, many architectures have been used to solve standard planes detection (e.g. SonoNet etc).

Thank you for your comment. InceptionV3, VGG16, and ResNet50 are widely recognized and extensively used benchmark classification architectures in the field of computer vision, including medical image analysis¹⁰. These models have been thoroughly studied and have consistently achieved state-of-the-art performance on a variety of medical image classification tasks. Conducting further experiments with other models such as DenseNet¹⁴, SENet¹⁵, or EfficientNet¹⁶ would require significant resources in terms of time and computational power. We chose to focus on these three models to ensure that our experiments were feasible within the available resources. Our approach was

data-centric, aiming to effectively leverage the available data and reduce the need for extremely large or computationally expensive models.

METHODS

MODELS AND TRAINING

It is unclear why the authors chose to use binary cross entropy as the loss function, considering that the problem involves multiple quality criteria, making it a potential multiclass problem. For instance, in the study, four quality criteria were assessed (kidneys not visible (A_KN), plane showing portal sinus (A_PS), plane showing stomach bubble (A_SB), and symmetrical plane (A_SYM)). It would be beneficial for the authors to explain the rationale behind using binary cross entropy in this context and consider an appropriate loss function for multiclass classification. How was the augmentation performed (e.g. on the fly / offline)? Additionally, it would be helpful if the authors specified the loss function used to train the Mask-RCNN model.

Thank you for your comment. The classification of standard planes based on multiple quality criteria is a multilabel rather than a multiclass classification problem, as each 2D ultrasound image may be assigned multiple quality criteria labels. The appropriate loss function for this type of problem is typically Binary Cross-Entropy Loss, as it treats each label independently and calculates the loss for each label separately. The objective is to optimize the model to output probabilities for each quality criterion, indicating the likelihood of that criterion being present. This approach facilitates a straightforward interpretation of the model's output, with each probability corresponding to a specific quality criterion.

Data augmentation was performed on the fly, as we have now clarified in the relevant paragraph: *"To prevent overfitting, we applied various data augmentation techniques on the fly using the following transformations: rotation between -15 and 15 degrees, zoom by 10%, brightness range between 0.2 and 0.8, as well as horizontal and vertical flipping."*

Regarding the loss function used for Mask-RCNN, it is the one defined in the original paper¹⁸. We have included the following sentence under the Models and Training subsection : *"The used loss function is similar to the one described in²⁴. It combines the classification loss, the bounding-box loss, and the average binary cross-entropy loss of the mask."*

Why did the authors decide to work with 2D images as opposed to videos?

Thanks for your comment. Only 2D ultrasound images were available for the retrospective portion of the study, as in the majority of radiology and obstetrics departments, scan data is saved as a set of images in the PACS system with few cine-loops available. However, in our opinion, training and testing a cluster of models on 2D images only does not translate to clinical use since the sonographer's workflow is, in its essence, dynamic. That is why we decided to add an extra step to our study and prospectively test the 2D-trained models on videos which few studies have done. We also imagined a way in which our approach would be useful to sonographers: lateral sweeps for AFV assessment and 'biometric' sweeps around biometry standard planes and tested our approach in that exact scenario.

To facilitate a fair comparison of the proposed methodology with state-of-the-art approaches, it is recommended that the authors include a reference to the HC18 challenge metrics for evaluating the metrics. This would provide a standardized framework for evaluating the performance of the proposed methodology.

Thank you for your comment. In addition to the AUC measure that we previously reported, we have added the F1 score metric to evaluate the performance of our INCEPTIONV3, RESNET50V2, and VGG16 classification models. The F1 scores for these models are 0.81, 0.80, and 0.80 for the abdomen, 0.66, 0.62, and 0.62 for the brain, and 0.81, 0.78, and 0.80 for the AF pocket, respectively. For our segmentation models, we have already reported the Dice scores, which is a commonly used metric for image segmentation and has been utilized in the HC18 challenge and reported in most of the state-of-the-art literature. Table 3 summarizes the performance results of our models alongside those of state-of-the-art models.

MINOR COMMENTS

Please provide better captions of figures.

Figure 1: What does "doctors, semi-automatic, total" mean? Is the classification performed only for the brain and abdomen? Why?

Thank you for this insightful comment. We have changed the captions to Figure 1, which now reads:

Figure 1: Summary of the retrospective data used during the segmentation and classification tasks along with the data amount used for training, validation, and testing. 'Doctors' refer to

physicians who prospectively and manually annotated standard planes. 'Semi-automatic' refers to the process of the standard plane and biometric measurement recognition using Optical Character Recognition, validated by a trained research technician.

Finally, we would like to express our sincere gratitude for your review of our scientific paper. Your valuable feedback and insights have helped us to improve our paper and make it even stronger. Thank you for taking the time to provide us with your thoughtful comments and for contributing to our work.

Bibliography

1. K. Rasheed, F. Junejo, A. Malik and M. Saqib, "Automated Fetal Head Classification and Segmentation Using Ultrasound Video," in *IEEE Access*, vol. 9, pp. 160249-160267, 2021, doi: 10.1109/ACCESS.2021.3131518.
2. Płotka, S. *et al.* Deep learning fetal ultrasound video model match human observers in biometric measurements. *Phys. Med. Biol.* (2022) doi:10.1088/1361-6560/ac4d85.
3. Fiorentino, M. C., Villani, F. P., Di Cosmo, M., Frontoni, E. & Moccia, S. A Review on Deep-Learning Algorithms for Fetal Ultrasound-Image Analysis. *arXiv:2201.12260 [cs, eess]* (2022).
4. Baumgartner, C. F. *et al.* SonoNet: Real-Time Detection and Localisation of Fetal Standard Scan Planes in Freehand Ultrasound. *IEEE Transactions on Medical Imaging* **36**, 2204–2215 (2017).
5. Cai, Y. *et al.* Spatio-temporal visual attention modelling of standard biometry plane-finding navigation. *Medical Image Analysis* **65**, 101762 (2020).
6. Zeng, Y., Tsui, P.-H., Wu, W., Zhou, Z. & Wu, S. Fetal Ultrasound Image Segmentation for Automatic Head Circumference Biometry Using Deeply Supervised Attention-Gated V-Net. *J Digit Imaging* **34**, 134–148 (2021).
7. Pokaprakarn, T. *et al.* AI Estimation of Gestational Age from Blind Ultrasound Sweeps in

- Low-Resource Settings. *NEJM Evidence* **0**, EVIDoa2100058.
8. Gomes, R. G. *et al.* A mobile-optimized artificial intelligence system for gestational age and fetal malpresentation assessment. *Commun Med* **2**, 1–9 (2022).
 9. Al-Bander, B., Alzahrani, T., Alzahrani, S., Williams, B. M. & Zheng, Y. Improving Fetal Head Contour Detection by Object Localisation with Deep Learning. in *Medical Image Understanding and Analysis* (eds. Zheng, Y., Williams, B. M. & Chen, K.) vol. 1065 142–150 (Springer International Publishing, 2020).
 10. Moccia, S., Fiorentino, M. C. & Frontoni, E. Mask-RCNN: a distance-field regression version of Mask-RCNN for fetal-head delineation in ultrasound images. *Int J CARS* **16**, 1711–1718 (2021).
 11. B. Al-Bander, T. Alzahrani, S. Alzahrani, B. M. Williams, and Y. Zheng, “Improving fetal head contour detection by object localization with deep learning,” in Annual Conference on Medical Image Understanding and Analysis. Springer, 2019, pp. 142–150.
 12. S. Moccia, M. C. Fiorentino, and E. Frontoni, “Maskr2CNN: a distance-field regression version of Mask-RCNN for fetal-head delineation in ultrasound images,” International Journal of Computer Assisted Radiology and Surgery, pp. 1–8, 2021.
 13. Zhao L, Li N, Tan G, Chen J, Li S, Duan M. The End-to-end Fetal Head Circumference Detection and Estimation in Ultrasound Images. *IEEE/ACM Trans Comput Biol Bioinform.* 2022 Dec 6;PP. doi: 10.1109/TCBB.2022.3227037. Epub ahead of print. PMID: 37015581.
 14. G. Huang, Z. Liu, L. Van Der Maaten, and K. Q. Weinberger, “Densely connected convolutional networks,” in 2017 IEEE Conference on Computer Vision and Pattern Recognition (CVPR), Honolulu, HI, USA, 2017.
 15. J. Hu, L. Shen, and G. Sun, “Squeeze-and-Excitation Networks,” in 2018 IEEE/CVF Conference on Computer Vision and Pattern Recognition (CVPR), Salt Lake City, Utah, USA, 2018.
 16. M. Tan, and Q. V. Le, “EfficientNet: Rethinking Model Scaling for Convolutional Neural

Networks,” in Proceedings of the 36th International Conference on Machine Learning, Long Beach, California, USA, 2019, pp. 6105– 6114.

REVIEWERS' COMMENTS

Reviewer #2 (Remarks to the Author):

The Authors have satisfactorily provided the answers and manuscript modifications in response to my critique. Thank you.

Reviewer #3 (Remarks to the Author):

Thank you for incorporating my comments and addressing the revisions in the manuscript. I appreciate your responsiveness and the thoroughness with which you have handled my questions. I would like to offer a suggestion regarding the citation of papers. Instead of citing "arXiv" papers, I would recommend citing the corresponding journal publication (e.g. Fiorentino et al "A review on deep-learning algorithms for fetal ultrasound-image analysis" published in Medical Image Analysis (p.102629)).

REBUTTAL LETTER

All changes in the revised manuscript have been highlighted in yellow.

Reviewer #2 (Remarks to the Author):

The Authors have satisfactorily provided the answers and manuscript modifications in response to my critique. Thank you.

Thank you for taking the time to review our manuscript and provide your valuable feedback. We are glad our responses and modifications have satisfactorily addressed your critique. We appreciate your contribution to improving the quality of our work.

Reviewer #3 (Remarks to the Author):

Thank you for incorporating my comments and addressing the revisions in the manuscript. I appreciate your responsiveness and the thoroughness with which you have handled my questions. I would like to offer a suggestion regarding the citation of papers. Instead of citing "arXiv" papers, I would recommend citing the corresponding journal publication (e.g. Fiorentino et al "A review on deep-learning algorithms for fetal ultrasound-image analysis" published in *Medical Image Analysis* (p.102629).

We are grateful for your insightful feedback and suggestion. Your time and effort in reviewing our manuscript have been invaluable in helping us improve it. In accordance with your suggestion, we have updated our citations to reference the corresponding journal publications instead of the "arXiv" papers. Thank you for bringing this to our attention.

41. **Fiorentino, M. C., Villani, F. P., Di Cosmo, M., Frontoni, E. & Moccia, S. A Review on Deep-Learning Algorithms for Fetal Ultrasound-Image Analysis. *Medical Image Analysis* **83**, 2023, 102629, ISSN 1361-8415.**

See below for the rest of the citations:

REFERENCES

1. Grytten, J., Skau, I., Sørensen, R. & Eskild, A. Does the Use of Diagnostic Technology Reduce Fetal Mortality? *Health Serv Res* **53**, 4437–4459 (2018).
2. Wiafe, Y., Odoi, A. & Dassah, E. The Role of Obstetric Ultrasound in Reducing Maternal and Perinatal Mortality. *in Ultrasound Imaging* (2011).
3. Carrera, J. M. Obstetric Ultrasounds in Africa: Is it Necessary to Promote their Appropriate Use? *Donald School Journal of Ultrasound in Obstetrics and Gynecology* **5**, 289–296 (2011).
4. Tunçalp, Ö., Pena-Rosas, J.P., Lawrie, T., Bucagu, M., Oladapo, O.T., Portela, A., Metin Gülmezoglu, A. WHO recommendations on antenatal care for a positive pregnancy experience—going beyond survival. *BJOG* 2017; 124: 860– 862.
5. Kim, E. T., Singh, K., Moran, A., Armbruster, D. & Kozuki, N. Obstetric ultrasound use in low and middle income countries: a narrative review. *Reprod Health* **15**, (2018).
6. Joseph KS, Boutin A, Lisonkova S, Muraca GM, Razaz N, John S, Mehrabadi A, Sabr Y, Ananth CV & Schisterman E. Maternal Mortality in the United States: Recent Trends, Current Status, and Future Considerations. *Obstet Gynecol.* 2021 May 1;137(5):763-771.
7. Melamed N, Baschat A, Yinon Y, Athanasiadis A, Mecacci F, Figueras F, Berghella V, Nazareth A, Tahlak M, McIntyre HD, Da Silva Costa F, Kihara AB, Hadar E, McAuliffe F, Hanson M, Ma RC, Gooden R, Sheiner E, Kapur A, Divakar H, Ayres-de-Campos D, Hirsch L, Poon LC, Kingdom J, Romero R & Hod M. FIGO (international Federation of Gynecology and obstetrics) initiative on fetal growth: best practice advice for screening, diagnosis, and management of fetal growth restriction. *Int J Gynaecol Obstet.* 2021 Mar;152 Suppl 1(Suppl 1):3-57.
8. Nardoza LM, Caetano AC, Zamarian AC, Mazzola JB, Silva CP, Marçal VM, Lobo TF,

- Peixoto AB & Araujo Júnior E. Fetal growth restriction: current knowledge. *Arch Gynecol Obstet.* 2017 May;295(5):1061-1077.
9. Lees, C.C., Stampalija, T., Baschat, A.A., da Silva Costa, F., Ferrazzi, E., Figueras, F., Hecher, K., Kingdom, J., Poon, L.C., Salomon, L.J. & Unterscheider, J. (2020), ISUOG Practice Guidelines: diagnosis and management of small-for-gestational-age fetus and fetal growth restriction. *Ultrasound Obstet Gynecol*, 56: 298-312.
 10. Morris RK, Meller CH, Tamblyn J, Malin GM, Riley RD, Kilby MD, Robson SC & Khan KS. Association and prediction of amniotic fluid measurements for adverse pregnancy outcome: systematic review and meta-analysis. *BJOG: An International Journal of Obstetrics & Gynaecology* **121**, 686–699 (2014).
 11. Yaqub M, Kelly B, Stobart H, Napolitano R, Noble JA & Papageorghiou AT. Quality-improvement program for ultrasound-based fetal anatomy screening using large-scale clinical audit. *Ultrasound Obstet Gynecol* **54**, 239–245 (2019).
 12. Kilani R, Aleyadeh W, Atieleh LA, Al Suleimat AM, Khadra M & Hawamdeh HM. Inter-observer variability in fetal biometric measurements. *Taiwanese Journal of Obstetrics and Gynecology* **57**, 32–39 (2018).
 13. Sande, J. A., Ioannou, C., Sarris, I., Ohuma, E. O. & Papageorghiou, A. T. Reproducibility of measuring amniotic fluid index and single deepest vertical pool throughout gestation. *Prenat Diagn* **35**, 434–439 (2015).
 14. Płotka, S., Klasa A., Lisowska, A., Seliga-Siwecka, J., Lipa, M., Trzciński, T. & Sitek, A.. Deep learning fetal ultrasound video model match human observers in biometric measurements. *Phys. Med. Biol.* **67**, 045013 (2022).
 15. Zeng, Y., Tsui, P.-H., Wu, W., Zhou, Z. & Wu, S. Fetal Ultrasound Image Segmentation

for Automatic Head Circumference Biometry Using Deeply Supervised Attention-Gated V-Net. *J Digit Imaging* **34**, 134–148 (2021).

16. Kim, H.P., Lee, S.M., Kwon, J.Y., Park, Y., Kim, K.C., Seo, J.K. Automatic evaluation of fetal head biometry from ultrasound images using machine learning. *Physiol Meas* **40**, 065009 (2019).
17. Burgos-Artizzu, X.P., Coronado-Gutiérrez, D., Valenzuela-Alcaraz, B. et al. Evaluation of deep convolutional neural networks for automatic classification of common maternal fetal ultrasound planes. *Sci Rep* **10**, 10200 (2020).
18. Pokaprakarn, T., Prieto, J.C., Price, J.T., Kasaro, M.P., Sindano, N., Shah, H.R., Peterson, M., Akapelwa, M.M., Kapilya, F.M., Sebastião, Y.V., Goodnight, W., Stringer, E.M., Freeman, B.L., Montoya, L.M., Chi, B.H., Rouse, D.J., Cole, S.R., Vwalika, B., Kosorok, M.R., Stringer, J.S.A. AI Estimation of Gestational Age from Blind Ultrasound Sweeps in Low-Resource Settings. *NEJM Evidence*, 2022 May;1(5):10.
19. Sendra-Balcells, C., Campello, V.M., Torrents-Barrena, J. et al. Generalisability of fetal ultrasound deep learning models to low-resource imaging settings in five African countries. *Sci Rep* **13**, 2728 (2023).
20. Gomes, R.G., Vwalika, B., Lee, C. et al. A mobile-optimized artificial intelligence system for gestational age and fetal malpresentation assessment. *Commun Med* **2**, 1–9 (2022).
21. Tkachenko, M., Malyuk, M., Holmanyuk, A., and Liubimov, N. (2020-2022). Label Studio: Data labeling software. Open source software available from <https://github.com/heartexlabs/label-studio>.
22. Salomon, L.J., Alfrevic, Z., Da Silva Costa, F., Deter, R.L., Figueras, F., Ghi, T., Glanc, P., Khalil, A., Lee, W., Napolitano, R., Papageorghiou, A., Sotiriadis, A., Stirnemann, J., Toi,

- A., Yeo, G. ISUOG Practice Guidelines: ultrasound assessment of fetal biometry and growth. *Ultrasound Obstet Gynecol.* 2019 Jun;53(6):715-723.
23. Salomon, L.J., Alfrevic, Z., Berghella, V., Bilardo, C., Hernandez-Andrade, E., Johnsen, S.L., Kalache, K., Leung, K.Y., Malinger, G., Munoz, H., Prefumo, F., Toi, A., Lee, W. Practice guidelines for performance of the routine mid-trimester fetal ultrasound scan. *Ultrasound in Obstetrics & Gynecology* **37**, 116–126 (2011).
24. He, K., Gkioxari, G., Dollár, P. & Girshick, R. Mask R-CNN. *2017 IEEE International Conference on Computer Vision (ICCV)* 2980–2988 (2017).
25. Al-Bander, B., Alzahrani, T., Alzahrani, S., Williams, B. M. & Zheng, Y. Improving Fetal Head Contour Detection by Object Localisation with Deep Learning. *Medical Image Understanding and Analysis* (eds. Zheng, Y., Williams, B. M. & Chen, K.) vol. 1065 142–150 (Springer International Publishing, 2020).
26. Moccia, S., Fiorentino, M. C. & Frontoni, E. Mask-R²CNN: a distance-field regression version of Mask-RCNN for fetal-head delineation in ultrasound images. *Int J CARS* **16**, 1711–1718 (2021).
27. Wu, Y., Kirillov, A., Massa, F., Lo, W.-Y., and Girshick, R. (2019). Detectron2. <https://github.com/facebookresearch/detectron2>.
28. Cho, H.C., Sun, S., Min Hyun C., Kwon, J.Y., Kim, B., Park, Y., Seo, J.K., Automated ultrasound assessment of amniotic fluid index using deep learning. *Medical Image Analysis* **69**, 101951 (2021).
29. Hadlock, F. P., Harrist, R. B., Sharman, R. S., Deter, R. L. & Park, S. K. Estimation of fetal weight with the use of head, body, and femur measurements—A prospective study. *American Journal of Obstetrics and Gynecology* **151**, 333–337 (1985).

30. Papageorghiou, A.T., Kemp, B., Stones, W., Ohuma, E.O., Kennedy, S.H., Purwar, M., Salomon, L.J., Altman, D.G., Noble, J.A., Bertino, E., Gravett, M.G., Pang, R., Cheikh Ismail, L., Barros, F.C., Lambert, A., Jaffer, Y.A., Victora, C.G., Bhutta, Z.A., Villar, J. Ultrasound-based gestational-age estimation in late pregnancy. *Ultrasound in Obstetrics & Gynecology* **48**, 719–726 (2016).
31. Espinoza, J., Good, S., Russell, E. & Lee, W. Does the use of automated fetal biometry improve clinical workflow efficiency? *J Ultrasound Med* **32**, 847–850 (2013).
32. Pels, A., Beune, I. M., van Wassenaer-Leemhuis, A. G., Limpens, J. & Ganzevoort, W. Early-onset fetal growth restriction: A systematic review on mortality and morbidity. *Acta Obstet Gynecol Scand* **99**, 153–166 (2020).
33. Figueroa, L., McClure, E.M., Swanson, J. et al. Oligohydramnios: a prospective study of fetal, neonatal and maternal outcomes in low-middle income countries. *Reproductive Health* **17**, 19 (2020).
34. Tashfeen, K. & Hamdi, I. M. Polyhydramnios as a Predictor of Adverse Pregnancy Outcomes. *Sultan Qaboos Univ Med J* **13**, 57–62 (2013).
35. Sarris, I., Ioannou, C., Chamberlain, P., Ohuma, E., Roseman, F., Hoch, L., Altman, D.G., Papageorghiou, A.T. Intra- and interobserver variability in fetal ultrasound measurements. *Ultrasound in Obstetrics & Gynecology* **39**, 266–273 (2012).
36. Perni, S.C., Chervenak, F.A., Kalish, R.B., Magherini-Rothe, S., Predanic, M., Streltsoff, J., Skupski, D.W. Intraobserver and interobserver reproducibility of fetal biometry. *Ultrasound in Obstetrics & Gynecology* **24**, 654–658 (2004).
37. Zhang, B., Liu, H., Luo, H. & Li, K. Automatic quality assessment for 2D fetal sonographic standard plane based on multitask learning. *Medicine (Baltimore)* **100**, e24427

(2021).

38. Wu L., Cheng, J.Z, Li, S., Lei, B., Wang, T. & Ni, D., FUIQA: Fetal Ultrasound Image Quality Assessment With Deep Convolutional Networks. *IEEE Transactions on Cybernetics* **47**, 1336–1349 (2017).
39. Hughes, D.S., Magann, E.F., Whittington, J.R., Wendel, M.P., Sandlin, A.T. & Ounpraseuth, S.T. Accuracy of the Ultrasound Estimate of the Amniotic Fluid Volume (Amniotic Fluid Index and Single Deepest Pocket) to Identify Actual Low, Normal, and High Amniotic Fluid Volumes as Determined by Quantile Regression. *Journal of Ultrasound in Medicine* **39**, 373–378 (2020).
40. Sun, S., Kwon, J. -Y., Park, Y., Cho, H.C., Hyun, C.M. & Seo J.K., Complementary Network for Accurate Amniotic Fluid Segmentation From Ultrasound Images. *IEEE Access* **9**, 108223–108235 (2021).
41. Fiorentino, M. C., Villani, F. P., Di Cosmo, M., Frontoni, E. & Moccia, S. A Review on Deep-Learning Algorithms for Fetal Ultrasound-Image Analysis. *Medical Image Analysis* **83**, 2023, 102629, ISSN 1361-8415.
42. Baumgartner, C. F. et al. SonoNet: Real-Time Detection and Localisation of Fetal Standard Scan Planes in Freehand Ultrasound. *IEEE Transactions on Medical Imaging* **36**, 2204–2215 (2017).
43. Cai, Y., Droste, R., Harshita, S., Chatelain, P., Drukker, L., Aris, T., Papageorghiou & J. Noble, A. Spatio-temporal visual attention modeling of standard biometry plane-finding navigation. *Medical Image Analysis* **65**, 101762 (2020).
44. Rasheed, K., Junejo, F., Malik, A. & Saqib, M. Automated Fetal Head Classification and Segmentation Using Ultrasound Video. *IEEE Access* **9**, 160249–160267 (2021).

45. Zhu, F., Liu, M., Wang, F., Qiu, D., Li, R., Dai, C. Automatic measurement of fetal femur length in ultrasound images: a comparison of random forest regression model and SegNet. *Math Biosci Eng* **18**, 7790–7805 (2021).
46. Li, P., Zhao, H., Liu, P. & Cao, F. Automated measurement network for accurate segmentation and parameter modification in fetal head ultrasound images. *Med Biol Eng Comput* **58**, 2879–2892 (2020).
47. Li, Y., Xu, R., Ohya, J. & Iwata, H. Automatic fetal body and amniotic fluid segmentation from fetal ultrasound images by encoder-decoder network with inner layers. *39th Annual International Conference of the IEEE Engineering in Medicine and Biology Society (EMBC)* 1485–1488 (2017).
48. Prieto, J.C., Shah, H., Rosenbaum, A.J., Jiang, X., Musonda, P., Price, J.T., Stringer, E.M., Vwalika, B., Stamilio, D.M., Stringer, J.S.A. An automated framework for image classification and segmentation of fetal ultrasound images for gestational age estimation. *Proc SPIE Int Soc Opt Eng* 11596, 115961N (2021).
49. Płotka, S., Włodarczyk, T., Klasa, A., Lipa, M., Sitek, A., Trzciński, T. (2021). FetalNet: Multi-task Deep Learning Framework for Fetal Ultrasound Biometric Measurements. In: Mantoro, T., Lee, M., Ayu, M.A., Wong, K.W., Hidayanto, A.N. (eds) *Neural Information Processing. ICONIP 2021. Communications in Computer and Information Science* **1517**. Springer, Cham.
50. Kehl, S., Schelkle, A., Thomas, A., Puhl, A., Meqdad, K., Tuschy, B., Berlit, S., Weiss, C., Bayer, C., Heimrich, J., Dammer, U., Raabe, E., Winkler, M., Faschingbauer, F., Beckmann, MW., Sütterlin, M. Single deepest vertical pocket or amniotic fluid index as evaluation test for predicting adverse pregnancy outcome (SAFE trial): a multicenter,

open-label, randomized controlled trial. *Ultrasound in Obstetrics & Gynecology* **47**, 674–679 (2016).